# AXL confers cell migration and invasion by hijacking a PEAK1-regulated focal adhesion protein network

Afnan Abu-Thuraia[1,2], Marie-Anne Goyette [1,2], Jonathan Boulais [1], Carine Delliaux [1], Chloé Apcher [1,2], Céline Schott[1,2], Rony Chidiac [3], Halil Bagci [1,4,10], Marie-Pier Thibault [1], Dominique Davidson[1], Mathieu Ferron [1,2,5], André Veillette [1,2], Roger J. Daly [6], Anne-Claude Gingras [7,8], Jean-Philippe Gratton[3] & Jean-François Côté [1,2,4,9 ✉]

Aberrant expression of receptor tyrosine kinase AXL is linked to metastasis. AXL can be activated by its ligand GAS6 or by other kinases, but the signaling pathways conferring its metastatic activity are unknown. Here, we define the AXL-regulated phosphoproteome in breast cancer cells. We reveal that AXL stimulates the phosphorylation of a network of focal adhesion (FA) proteins, culminating in faster FA disassembly. Mechanistically, AXL phosphorylates NEDD9, leading to its binding to CRKII which in turn associates with and orchestrates the phosphorylation of the pseudo-kinase PEAK1. We find that PEAK1 is in complex with the tyrosine kinase CSK to mediate the phosphorylation of PAXILLIN. Uncoupling of PEAK1 from AXL signaling decreases metastasis in vivo, but not tumor growth. Our results uncover a contribution of AXL signaling to FA dynamics, reveal a long sought-after mechanism underlying AXL metastatic activity, and identify PEAK1 as a therapeutic target in AXL positive tumors.

[1] Montreal Clinical Research Institute (IRCM), Montréal, QC H2W 1R7, Canada. [2] Molecular Biology Programs, Université de Montréal, Montréal, QC H3T 1J4, Canada. [3] Department of Pharmacology and Physiology, Université de Montréal, Montréal, QC H3C 3J7, Canada. [4] Department of Anatomy and Cell Biology, McGill University, Montréal, QC H3A 0C7, Canada. [5] Division of Experimental Medicine, McGill University, Montréal, QC H4A 3J1, Canada. [6] Cancer Program, Biomedicine Discovery Institute and Department of Biochemistry and Molecular Biology, Monash University, Clayton, VIC 3800, Australia. [7] Lunenfeld-Tanenbaum Research Institute, Sinai Health System, Toronto, ON M5G 1X5, Canada. [8] Department of Molecular Genetics, University of Toronto, Toronto, ON M5S 1A8, Canada. [9] Department of Biochemistry and Molecular Medicine, Université de Montréal, Montréal, QC H3C 3J7, Canada. [10]Present address: Institute of Biochemistry, ETH Zürich, Otto-Stern-Weg 3, 8093 Zürich, Switzerland. ✉email: jean-francois.cote@ircm.qc.ca

Tumor metastasis is the major cause of death of patients afflicted with breast cancer[1]. Successful treatment of metastatic breast cancers is currently a major clinical challenge in solid tumor oncology. Metastasis is a complex process that involves the invasion of tumor cells from the primary site into the surrounding tissue, intravasation into the circulation, and extravasation into secondary sites[1]. Among the breast cancer subtypes, human epidermal growth factor receptor 2 (HER2) positive and triple negative (triple-negative breast cancer (TNBC)) are characteristically aggressive, prone to metastasize, and patients show increased recurrence and lower rates of survival than with other subtypes[2]. Because TNBC lack the expression of estrogen receptor, progesterone receptor, and HER2, they do not respond to the corresponding targeted therapies and are typically treated by less effective standard chemotherapeutic regimens[3,4]. A better understanding of the molecular mechanisms that promote metastasis could help to develop new anti-metastatic therapies to improve outcomes in the treatment of TNBC.

The evolution of breast cancer tumors toward metastasis involves the re-activation of the developmental epithelial-to-mesenchymal (EMT) program that increases cell migration and invasion[5]. TNBC cells display several such EMT features. The TAM (TYRO3, AXL, MERTK) receptor tyrosine kinases (RTKs) form a distinct group that is activated by the atypical vitamin K-dependent and γ-carboxylated ligands growth-arrest-specific protein 6 (GAS6) and protein S (PROS)[6,7]. TAMs exert their functions in several biological processes, such as dampening the immune response and clearing apoptotic cells[8]. Among TAMs, AXL expression is associated with metastasis in solid cancers of several origins and correlates with poor patient survival[9]. Despite its overexpression in almost all TNBC cell lines, studies have shown AXL expression to be subtype independent in patients' breast tumors[10,11]. AXL is not expressed in the bulk of all breast tumors, but recent work suggests that its expression is in the cells that acquire EMT features[11]. AXL is a particularly attractive therapeutic target in metastatic cancers since it is involved in EMT and is positively correlated with chemo-resistance and targeted drug resistance[12–14]. AXL activation in epithelial cancers is complex and can occur by crosstalk with other RTKs as well as ligand binding. For example, AXL can be trans-activated by HER2 and can function in a GAS6-independent manner to promote metastasis[11]. However, in marked contrast with RTKs involved in cancer progression such as epidermal growth factor receptor (EGFR) and mesenchymal-to-epithelial transition, little is known about the specific mechanisms induced upon AXL activation to promote tumor invasiveness, metastasis, and other features, such as drug resistance.

By using quantitative phosphoproteomics to define the signaling pathways and biological processes that are impacted when AXL is activated by GAS6 in a TNBC cell model, we have identified a major contribution of AXL to the regulation of focal adhesion (FA) dynamics. We report a signaling pathway downstream of AXL activation that implicates the pseudo-kinase PEAK1 in coordinating FA turnover through recruitment of the canonical FA turnover module composed of β-PIX/GIT1/PAK to PAXILLIN (PXN). CRISPR/CAS9-mediated gene targeting of PEAK1 generated PEAK1-deficient cells that show impaired metastasis in vivo that can be rescued by PEAK1 wild type (WT) and not by a PEAK1 mutant that is uncoupled from AXL signaling. These results reveal a previously unknown contribution of AXL to the dynamics of FAs and expose opportunities to limit AXL-driven cell invasion.

## Results

**Defining the GAS6-induced AXL phosphoproteome.** To define the AXL signaling mechanism, we identified components of the AXL-regulated phosphoproteome using quantitative mass spectrometry (MS) of cellular extracts derived from stable isotope labeling of amino acids in cell culture (SILAC)[15]. More specifically, we compared the phosphoproteome derived from the invasive TNBC cell line Hs578T grown in the presence of either heavy or light isotope-labeled amino acids to identify signaling proteins that respond to AXL stimulation (Fig. 1a). These cells were used as model due to their high expression of AXL and not the related TAM family members TYRO3 or MERTK[16]. To activate AXL, we produced the recombinant AXL ligand GAS6 (GAS6-His) in engineered tissue culture cells (Supplementary Fig. 1a). Notably, γ-carboxylation of GAS6 is required for activation of AXL[6,17], and our recombinant GAS6 was found to contain the correct post-translational modification in our conditioned media as verified using a γ-carboxylation-specific anti-Gla antibody[18] (Supplementary Fig. 1b). The γ-carboxylation was also shown to be dependent on the presence of Vitamin K, as expected, and was abolished by warfarin treatment (Supplementary Fig. 1b). The GAS6 concentrations routinely attained in our conditioned media varied between 200 and 300 ng/mL (Supplementary Fig. 1c). We determined that the concentration of GAS6 required to obtain optimal AXL activation was 26 ng/mL, and this concentration was subsequently used to treat serum-starved heavy isotope-labeled cells for 5, 10, or 20 min while the light isotope-labeled cells were treated with control conditioned medium (Fig. 1a). These GAS-His treatments promoted AXL auto-phosphorylation and activation of its downstream target AKT (Fig. 1b).

GAS6 has been shown to bind to negatively charged phospholipids, such as phosphatidylserine exposed on apoptotic cells, to activate AXL[19]. To ensure that the AXL activation we observed using our conditioned medium was not due to GAS6 complexed to apoptotic debris, we quantified the percentage of apoptotic cells present in the conditioned media of the GAS6-His cells as well as in the target Hs578T cells (Supplementary Fig. 1d, e). By performing Annexin V-fluorescein isothiocyanate (FITC) and propidium iodide (PI) staining, neither the conditioned media of GAS6-His cells (either treated or not with tetracycline) nor the serum-starvation conditions in Hs578T cells led to an increase in the percentage of apoptotic cells. This suggests that the AXL activation observed following the treatment of Hs578T cells with GAS6 conditioned media is largely due to GAS6-His binding to AXL and is not dependent on apoptotic cell debris in the experiment. We further coated two different concentrations of liposomes with soluble GAS6 and tested their ability to activate AXL (Supplementary Fig. 1f) and found that the GAS6-coated liposomes promoted AKT phosphorylation in a similar manner to GAS6-His conditioned media. This suggests that GAS6-His is likely coated with phosphatidylserine vesicles in the conditioned media or that the Hs578T cells exposed some phosphatidylserine at their surface in a physiological manner. These data indicate that our GAS6-His conditioned medium is a valid tool to study the activation of AXL and downstream signaling.

To map the global phosphorylation events modulated by AXL activation, we performed phosphopeptide enrichment using TiO₂ chromatography and pY100 immunoaffinity on the mixed lysates of non-treated and GAS6-treated cells (Fig. 1a, b; Supplementary Fig. 2a). High-resolution liquid chromatography–tandem MS (LC-MS/MS) was employed to measure the relative abundance of phosphopeptides enriched upon GAS6-induced AXL activation. Owing to the short treatment times, protein abundance was assumed to be constant. By combining data from all time points, we quantified 5065 unique phosphopeptides (in a total of 2059 proteins), among which a curtailed list of 701 phosphoproteins was found to be modulated at least 1.5-fold by GAS6 across the three time points (Fig. 1c; Supplementary Data 1 and Supplementary Fig. 2b–e). Among those, we identified and validated

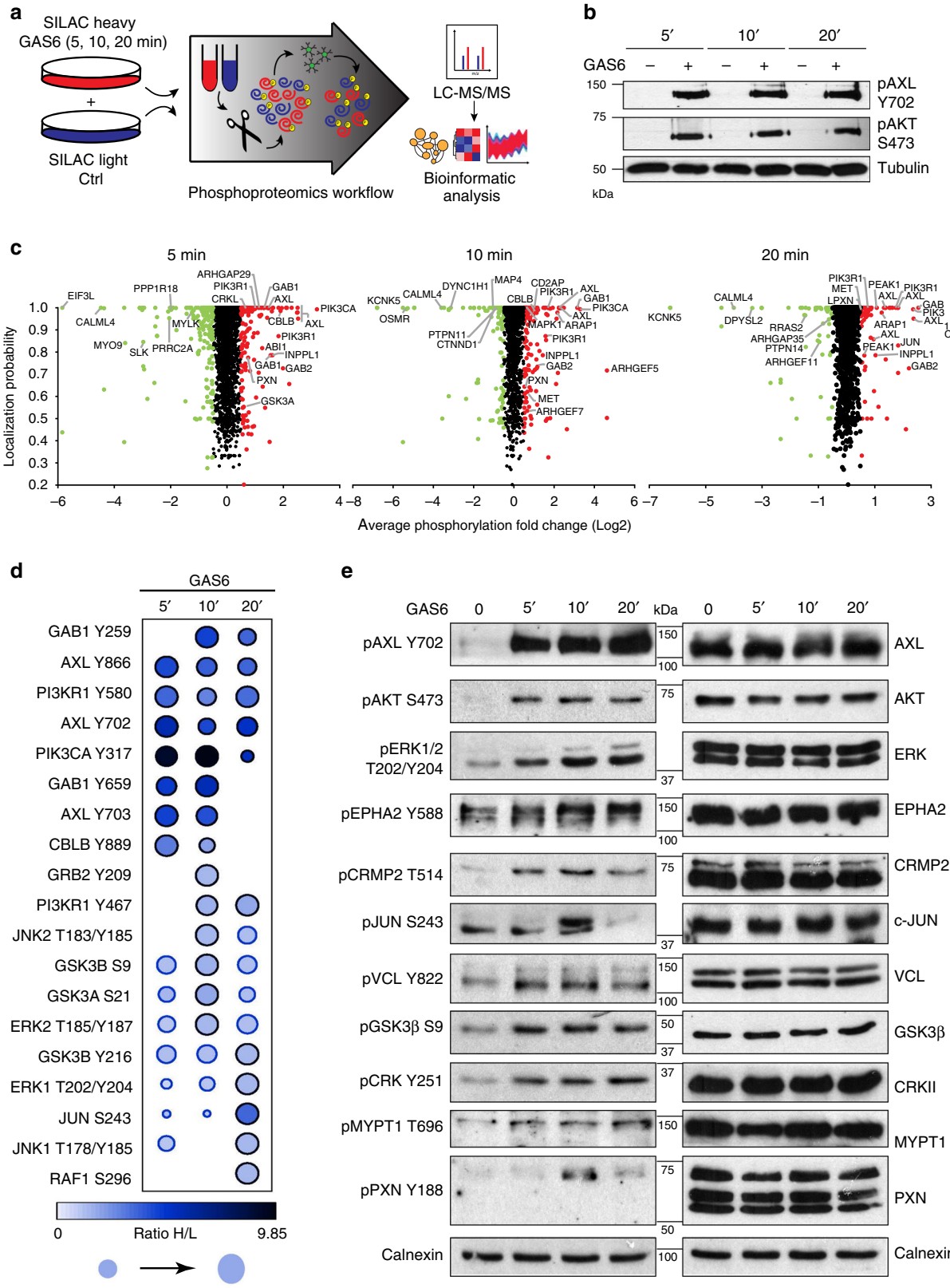

general targets of RTK pathways, including c-JUN, glycogen synthase kinase 3β (GSK3β), extracellular signal–regulated kinase, PI3KCA, c-Jun N-terminal kinase 1 (JNK1), JNK2, GSK3α, and RAF (Fig. 1d, e). The phosphorylation patterns obtained by western blots mostly reflected the SILAC ratios obtained in our phosphoproteomic dataset (Supplementary Data 1), confirming

that our screens reveal signaling pathways modulated following AXL activation.

**Overview map of signaling pathways regulated by AXL.** Use of the Kyoto Encyclopedia of Genes and Genomes (KEGG) pathway

**Fig. 1 Phosphoproteomic analyses of the receptor tyrosine kinase AXL in TNBC model. a** Schematic workflow of Hs578T cell labeling, treatment, and sample preparation for phosphopeptides enrichment and phosphoproteomic analyses. **b** Immunoblot analysis of SILAC-labeled Hs578T cell lysates collected at different time points demonstrates the phosphorylation of AXL and AKT following GAS6 stimulation. **c** Graphs of differential phosphopeptide abundances and phosphorylation probabilities in GAS6-treated cells vs. unstimulated cells. Phosphosites are deemed modulated if they exhibit SILAC ratios cutoffs of 1.5-fold increase or decrease, which is a phospho-modulation at a Log2 fold change ≥0.5 (phosphorylated—red circles) or ≤0.5 (dephosphorylated—green circles). **d** Dot plot representation of canonical phosphorylation sites regulated downstream of RTKs that were identified in our screen. Color of circle represents the phosphorylation level (ratio H/L) at 5-, 10- and 20-min. Border color of the circle depicts the significance of the modulation, whereas the size of the node indicates the relative abundance. The average of two ratios (H/L) was used to generate the relative abundance between different time points. **e** Lysates of Hs578T cells treated with GAS6 at three different time points were analyzed by immunoblotting for the phosphorylation status of known targets of receptor tyrosine kinase signaling.

database to analyze the GAS6-modulated phosphoproteins revealed significant enrichment of pathways known to be regulated by AXL activity (Fig. 2a; Supplementary Fig. 2f). We further generated a protein interactome based on the phosphoproteomic data and subdivided the list of proteins into subnetworks that were significantly enriched following AXL activation according to biological function (Fig. 2b, Supplementary Data 2). In agreement with the established roles of AXL, we identified modulation in pathways including "phagocytosis" and "mTOR signaling." Notably, we also identified multiple pathways not previously linked to AXL signaling including "focal adhesion," "RNA transport," and "Rap1 signaling" (Fig. 2b, Supplementary Data 2).

We focused on proteins involved in FA dynamics and regulation of the actin cytoskeleton as they were found to be the most phospho-modulated proteins by AXL and could be strong candidates in providing mechanistic insights for AXL's role in promoting cell migration, invasion, and metastasis. To determine whether FA dynamics are preferentially modulated by AXL in comparison to other RTKs, we compared our AXL phosphoproteomic dataset with available EGFR phosphoproteomic datasets. Interestingly, when comparing to EGFR datasets previously derived in HeLa cells[20,21], we defined a set of 331 unique and 195 shared phospho-modulated proteins (Fig. 2c). The unique AXL phospho-modulated proteins were significantly enriched for FA proteins, whereas EGFR preferentially modulated proteins were involved in adherens junctions (Fig. 2d). Hence, these data reveal that AXL, in contrast to EGFR, preferentially and robustly modulates the phosphorylation of FA proteins and provide insight into the unique phosphoproteome modulated by AXL activation.

**AXL promotes FA turnover.** The specific enrichment of FA proteins among the targets of AXL signaling prompted us to investigate whether AXL itself is localized at FA sites in TNBC cells. Using a proximity ligation assay (PLA) and the cytoskeletal protein PXN as a marker for FAs, a pool of AXL was indeed found to localize at PXN FAs, which was significantly decreased when cells were treated with the AXL inhibitor R428[22] (Fig. 3a, b). We further quantified the number of PXN FAs following modulation of AXL kinase activity with R428, GAS6, GAS6 and R428 or when its expression level was knocked down by small interfering RNA (siRNA) in MDA-MB-231 or Hs578T cells. In a motile cell, FA turnover is activated, leading to less stable adhesions. In contrast, serum starvation leads to a decrease in cell motility and cells tend to have a high number of stable adhesions due to their slow turnover. Interestingly, we found AXL that activation by GAS6 treatment of serum-starved cells led to a decrease in the number of FAs, whereas inhibiting its activity with R428 or decreasing its expression via siRNA knockdown in serum-containing conditions led to an increase in the FA number (Fig. 3c–e; Supplementary Fig. 3a–c). In addition, inhibiting AXL activity with R428 in GAS6-treated serum-starved cells reversed the effects of GAS6 on FA numbers. To determine whether AXL

regulates FA turnover, we analyzed FA lifetime, assembly, and disassembly times with a live-cell imaging approach of MDA-MB-231 expressing green fluorescent protein (GFP)-tagged PXN as an FA marker. We found that, in contrast to AXL inhibition or a decrease in *AXL* expression, an increase in AXL activity led to a decrease in the lifetime and the disassembly time of FAs without affecting the FA assembly time (Fig. 3f–h, Supplementary Videos 1 and 2). Similar results were also obtained using Hs578T cells (Supplementary Fig. 3d, e). These results suggest that the dynamics of FAs in TNBC are regulated by AXL catalytic activity.

**NEDD9 is an AXL substrate mediating FA signaling and invasion.** We hypothesized that scaffold proteins that become tyrosine phosphorylated in response to GAS6 might connect AXL signaling to changes in FA dynamics. Our phosphoproteomics dataset revealed that PI3KR1 and PI3KCA (PI3K subunits), GAB1/2 (PI3K and MAPK signaling), STAM1/2 (endosome trafficking) and the CAS family proteins (p130CAS, NEDD9 and CASS4), all of which have previously been implicated in the regulation of actin and FA signaling, are modulated in response to GAS6. Of those, NEDD9 was of interest as a candidate to transmit AXL signals to FAs since it is a scaffold protein downstream of integrin signaling and it has been implicated in migration and metastasis[23,24]. *AXL* downregulation by siRNA in MDA-MB-231 cells resulted in decreased tyrosine phosphorylation on NEDD9, whereas GAS6 treatment increased it (Fig. 4a, b), validating our phosphoproteomic data. To test whether NEDD9 could be a direct substrate of AXL, we carried out in vitro kinase assays using purified AXL (WT or kinase dead) and either recombinant NEDD9 substrate domain (SD) or C-terminal domain (CT) as substrates (Supplementary Fig. 4a). AXL WT, but not kinase dead, phosphorylates NEDD9 on its substrate domain and addition of an AXL inhibitor, but not SRC inhibitor, prevented this phosphorylation (Supplementary Fig. 4b). The total tyrosine phosphorylation of NEDD9 was regulated similarly in co-transfection assays (Supplementary Fig. 4c). These results reveal NEDD9 as a specific AXL substrate.

We next validated whether AXL promotes canonical NEDD9 signaling. NEDD9 is known to associate with the adapter protein CRKII to act as a molecular switch to activate RAC1 via the guanine nucleotide exchange factor DOCK3[23]. Because the tyrosine on NEDD9 whose phosphorylation is regulated by AXL (pY[241]DFP) falls in the pYXXP consensus for interaction with CRK proteins, we tested whether NEDD9 phosphorylation by AXL might regulate complex formation with CRKII. Knockdown of *AXL* by siRNA in MDA-MB-231 cells led to a decrease in NEDD9 phosphorylation and a decrease in CRKII binding to NEDD9 (Fig. 4c). We also confirmed the role of NEDD9 in promoting CRKII/DOCK3-induced RAC-mediated migration and invasion in TNBC cells (Supplementary Fig. 4d–g). NEDD9 localizes to FAs in part through association with the focal adhesion kinase (FAK)[25]. To investigate whether AXL controls

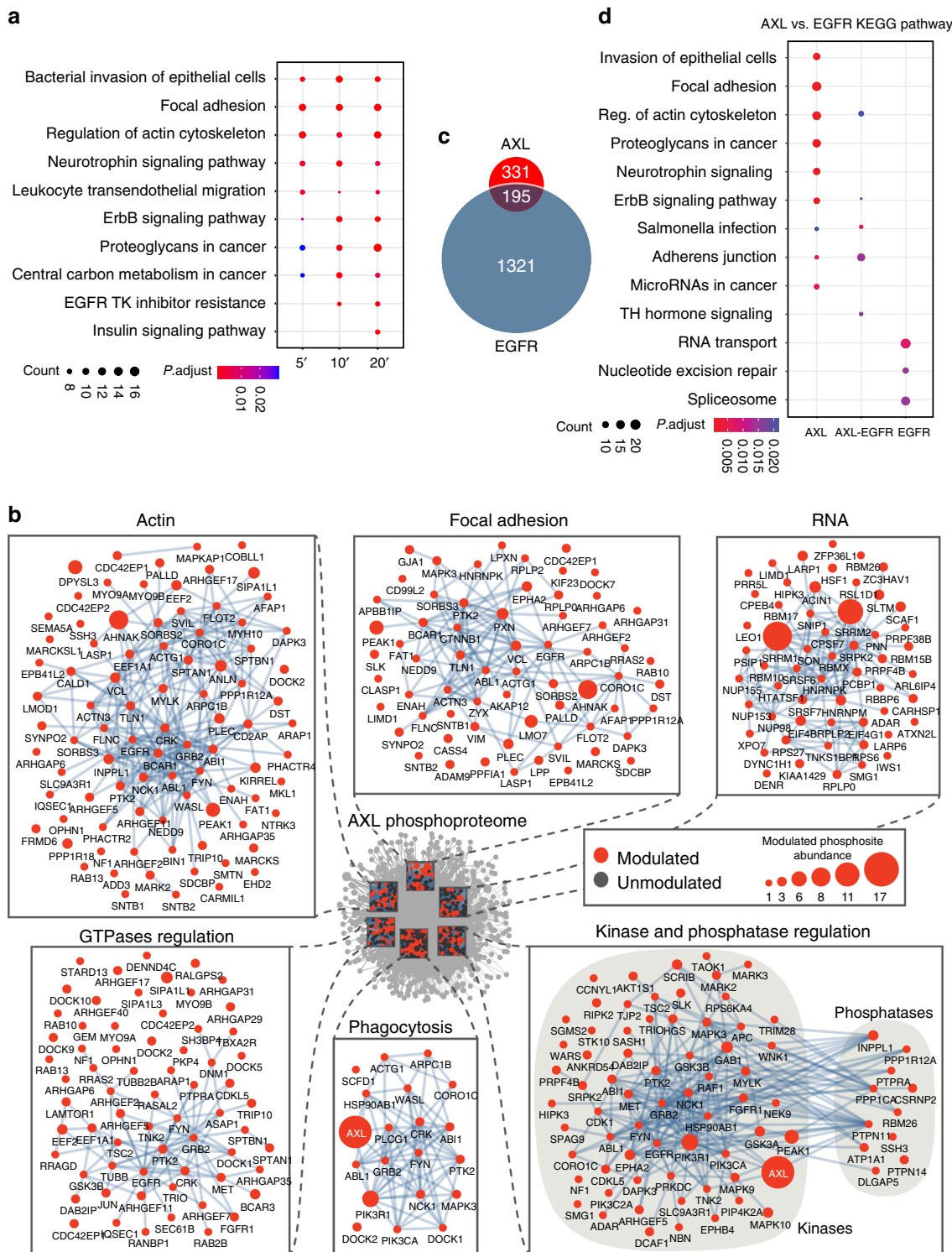

**Fig. 2 High-resolution overview of AXL signaling revealed in TNBC cells. a** Dot plot representation of top ten significantly enriched KEGG pathways of GAS6-regulated phosphoproteins at three different time points of stimulation. Circle sizes represent the number of regulated phosphoproteins associated with the specific pathway, and the color of the circle represents the significant adjusted *P* value. **b** Protein–protein interaction network analysis of GAS6-modulated (red nodes) and unmodulated phosphoproteins (black nodes). Surrounding subnetworks in zoom boxes highlight the selected and relevant functions of modulated phosphoproteins. Node sizes represent the number of significantly modulated phosphosites. **c** A Venn diagram comparing the number of AXL phospho-modulated proteins detected vs. the EGFR phospho-modulated proteins. **d** Dot plot representation of the significantly enriched KEGG pathways of GAS6-regulated phosphoproteins and EGF-regulated phosphoproteins. Circle sizes represent the number of regulated phosphoproteins associated with the specific pathway, and the color of the circle represents the significant adjusted *P* value.

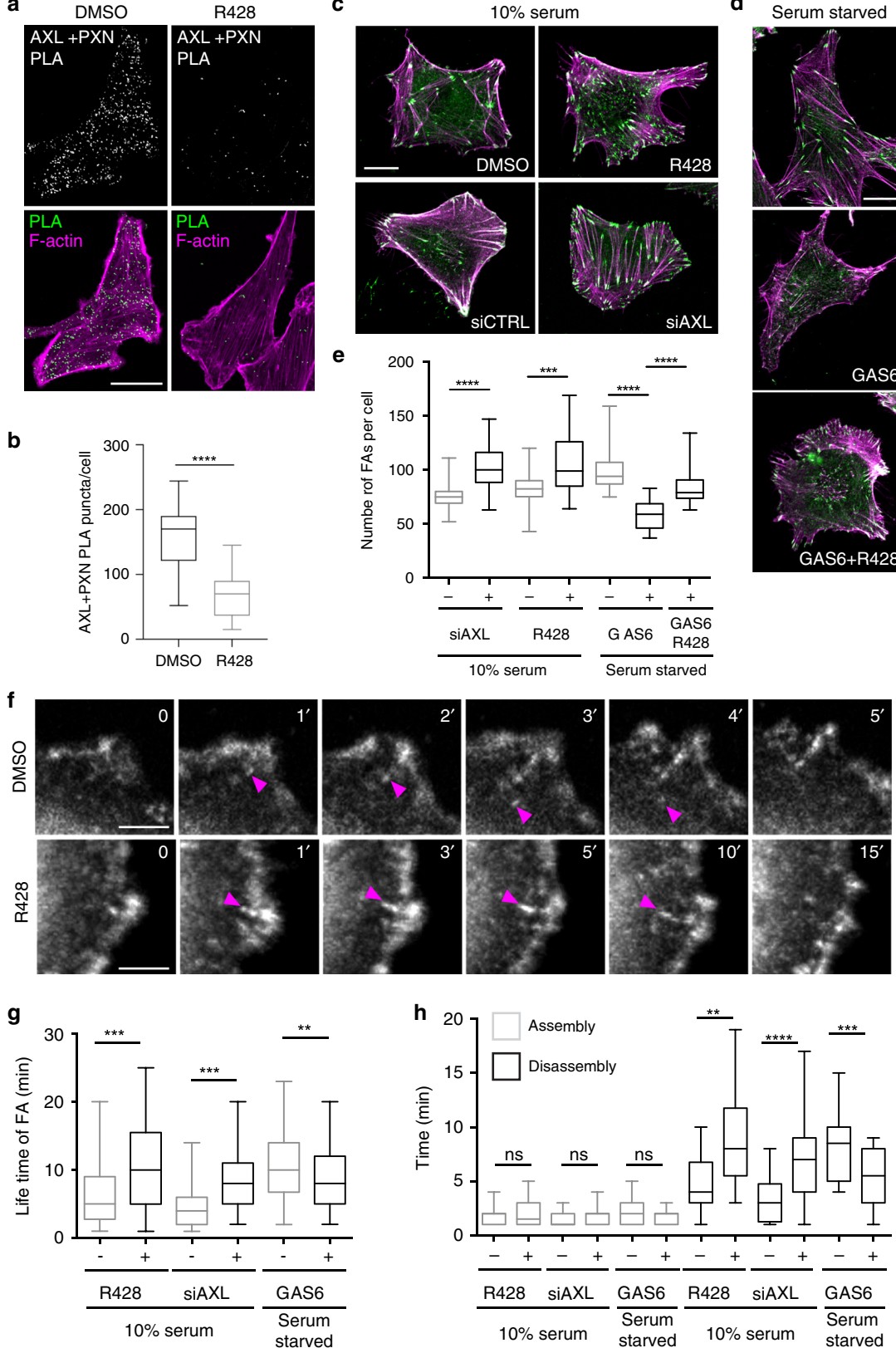

NEDD9 localization at FAs, Hs578T cells expressing GFP-NEDD9 were treated with AXL inhibitor R428 or transfected with siAXL. Using FAK as an FA marker, we found that either inhibiting AXL activity or decreasing its expression levels by siRNA led to an enrichment of NEDD9 at FAK FAs (Fig. 4d, e).

We also noted an increase in NEDD9/FAK complex formation upon WT but not kinase-dead AXL overexpression (Fig. 4f). Collectively, these results identify NEDD9 as a direct AXL substrate that regulates its location to FA sites for RAC1-induced cell migration.

**Fig. 3 AXL localizes at focal adhesion (FA) sites and modulates their turnover. a** AXL localizes at FA sites. Representative confocal images of MDA-MB-231 cells treated either with DMSO or 1 μM R428. Proximity ligation assay (PLA) was used to analyze localization of AXL at PXN-positive FAs. Scale bar, 20 μm. **b** Quantifications of the number of PLA puncta per cell per condition. ****$P < 0.0001$. ($n = 3$ experiments, 15 cells per condition per experiment). **c**, **d** AXL modulates the number of FAs per cell. Representative confocal images of Hs578T cells treated with DMSO or 1 μM R428, transfected with 100 nM of siCTRL or siAXL (**c**), serum starved and treated with GAS6 for 20 min, or serum starved and treated with GAS6 and R428 (**d**). Cells were stained for PXN (green) and Phalloidin (magenta). Scale bar, 10 μm. **e** Quantification of the FA number per cell depicted in **c**, **d**. ***$P = 0.0001$, ****$P < 0.0001$ ($n = 3$ experiments with 30 cells per condition). **f** MDA-MB-231 GFP-PXN-expressing cells treated with DMSO or 1 μM R428 were imaged live by spinning disk microscopy for a period of 30 min to assess the dynamics of FA turnover. Magenta arrow heads point toward a FA that was followed for its assembly and disassembly times. Scale bar, 5 μm. **g**, **h** Quantification of FA lifetime (**g**) and their assembly and disassembly times (**h**) of cells depicted in **f** and MDA-MB-231 GFP-PXN-expressing cells treated with GAS6 for 30 min or transfected with 100 nM siAXL and imaged as mentioned in **f**. ***$P < 0.0001$ (**g**), **$P = 0.00899$ (**g**), **$P = 0.000014$ (**h**), ***$P = 0.00009$ (**h**), ****$P = 0.000009$ (**h**) ($n = 3$ experiments with 90 FAs followed per condition). Data are represented as boxplots where the middle line is the median, the lower and upper hinges correspond to the first and third quartiles, the upper whisker extends from the hinge to the largest value, and the lower whisker extends from the hinge to the smallest value (**b**, **e**, **g**, **h**). Two-tailed unpaired $t$ test was used. Source data are provided as a Source data file.

**AXL recruits PEAK1 to FAs in proximity to NEDD9.** Next, we investigated how NEDD9 may contribute to FA dynamics downstream of AXL. To identify NEDD9 protein complexes, we used biotin identification (BioID), a biotin labeling technique coupled to MS, to capture proteins that either interact or are proximal to an engineered BirA*-FLAG-NEDD9 "bait" protein expressed in living cells[26] (Fig. 4g, Supplementary Fig. 4h). To do this, we generated Flp-In T-REx 293 cells so that BirA*-FLAG-NEDD9 could be expressed in a tetracycline-inducible manner. After addition of biotin to cells, leading to biotinylation of BirA*-FLAG-NEDD9, 133 high-confidence proximal interactors were identified. Our approach was validated by the fact that we identified ten known NEDD9 interactors (Supplementary Fig. 4i and Data 3), and it also identified several NEDD9 proximal partners. NEDD9 proximal proteins were clustered based on their gene ontology in corresponding biological processes, which provided further support of NEDD9 as a candidate for regulation of FA signaling (Fig. 4h, Supplementary Data 4).

To uncover the molecular mechanism whereby AXL mediated FA dynamics, we intersected the AXL phosphoproteomic dataset with the NEDD9 proximal interactors, and this revealed PEAK1 as a previously unknown proximity partner to NEDD9 and a protein that is phosphotyrosine-modulated by AXL (Fig. 4h, Supplementary Data 1 and 3). PEAK1 is a pseudo-kinase that has previously been linked to cell migration and FA turnover[27–29], but how it achieves these functions is unresolved. We detected by both PLA and co-immunoprecipitation (co-IP) an interaction of PEAK1 with AXL, which led to its AXL-mediated phosphorylation (Fig. 5a–c). Our MS data revealed five tyrosine phosphorylated sites with Y531 being predicted by PhophoNET as a top candidate site. However, this phosphorylation likely occurs on multiple tyrosine residues since a PEAK1[Y531F] mutant did not show a decrease in its global tyrosine phosphorylation by AXL (Supplementary Fig. 5a). To determine whether AXL kinase activity controls PEAK1 localization at PXN FAs, we treated MDA-MB-231 cells with R428, GAS6, or R428 and GAS6. We found that, upon AXL activation with GAS6, PEAK1 localization at PXN FAs was increased, whereas inhibiting AXL kinase activity with R428 decreased this localization (Fig. 5d, e). In addition, inhibiting AXL activity with R428 in GAS6-treated serum-starved cells reversed the effects of GAS6 on PEAK1 localization at PXN FAs (Fig. 5d, e). Similarly, inhibiting AXL kinase activity led to a redistribution of PEAK1 from the cell periphery to a more cytoplasmic distribution (Fig. 5f, g). These data demonstrate a role for PEAK1 as a candidate to mediate AXL signaling to FAs.

**CRKII binds PEAK1 to mediate AXL signaling to FAs.** BioID of NEDD9 revealed a proximal interaction with PEAK1 that could not be detected by co-IP. This raised the possibility that

they have a common membership to a protein complex and a bridging molecule is needed. We next defined the BirA*-FLAG-PEAK1 interactome by BioID proteomics to better understand how PEAK1 communicates with the FA machinery. The intersection of the AXL phosphoproteomic dataset with the PEAK1 BioID screen revealed a major proximal interactor, CRKII, a binding partner of NEDD9, that is also phospho-modulated by AXL activation (Fig. 5h; Supplementary Fig. 5b, Supplementary Data 5 and 6). Since CRK adapter proteins have been reported to assist in FA turnover[30] and emerged from these analyses, we further investigated CRKII role in PEAK1 regulation of FA.

PEAK1 contains a putative proline-rich motif that fits the consensus for CRK binding[31,32]. To test whether this is involved in their interaction, we generated a mutant of PEAK1 in which this proline-rich region was disrupted (PEAK1[3PA]) and we demonstrated that this nearly eliminated PEAK1 binding to CRKII by both co-IP and glutathione S-transferase (GST) pulldown assays (Fig. 6a, b; Supplementary Fig. 5c). Further analysis revealed that PEAK1/CRKII coupling occurred via the CRKII middle SH3 domain (Supplementary Fig. 5d). Indeed, comparison of the PEAK1[WT] and PEAK1[3PA] BioID interactomes revealed that the PEAK1[3PA] mutant no longer bound the CRK adapter proteins (Supplementary Data 5).

We also found that AXL-mediated phosphorylation of PEAK1 was abolished upon mutation of the CRKII-binding domain of PEAK1, despite its correct localization in the cell, suggesting that PEAK1 phosphorylation downstream of AXL requires its coupling to CRKII (Fig. 6c; Supplementary Fig. 5e). This was further demonstrated by co-IP of CRKII with PEAK1 in conditions only where AXL WT was expressed and not AXL kinase dead (Fig. 6d). In addition, localization of CRKII at FAK FAs, assessed by PLA, was diminished upon knockdown of PEAK1 in MDA-MB-231 cells, suggesting that the PEAK1/CRKII interaction is necessary for CRKII localization at FAs (Fig. 6e, f). CRKII complex formation with PXN is known to be essential for increased FA turnover and the induction of cell migration[33]. Hence, we examined whether CRKII recruitment to PXN is mediated by AXL. Indeed, co-IPs revealed that CRKII interaction with PXN was induced upon AXL overexpression and was dependent on AXL kinase activity (Fig. 6g).

**PEAK1 orchestrates the phosphorylation of PXN via CSK.** Accumulation of PXN phosphorylation at the sites of FAs is an indicator of FA turnover. Since PEAK1 and PRAG1 are related pseudo-kinases that associate with tyrosine kinases when phosphorylated[34], we tested whether PEAK1 might control the phosphorylation of PXN. Indeed, PEAK1 knockdown using siRNA eliminated PXN phosphorylation in TNBC cells (Fig. 7a) and an increase in PXN phosphorylation was observed following

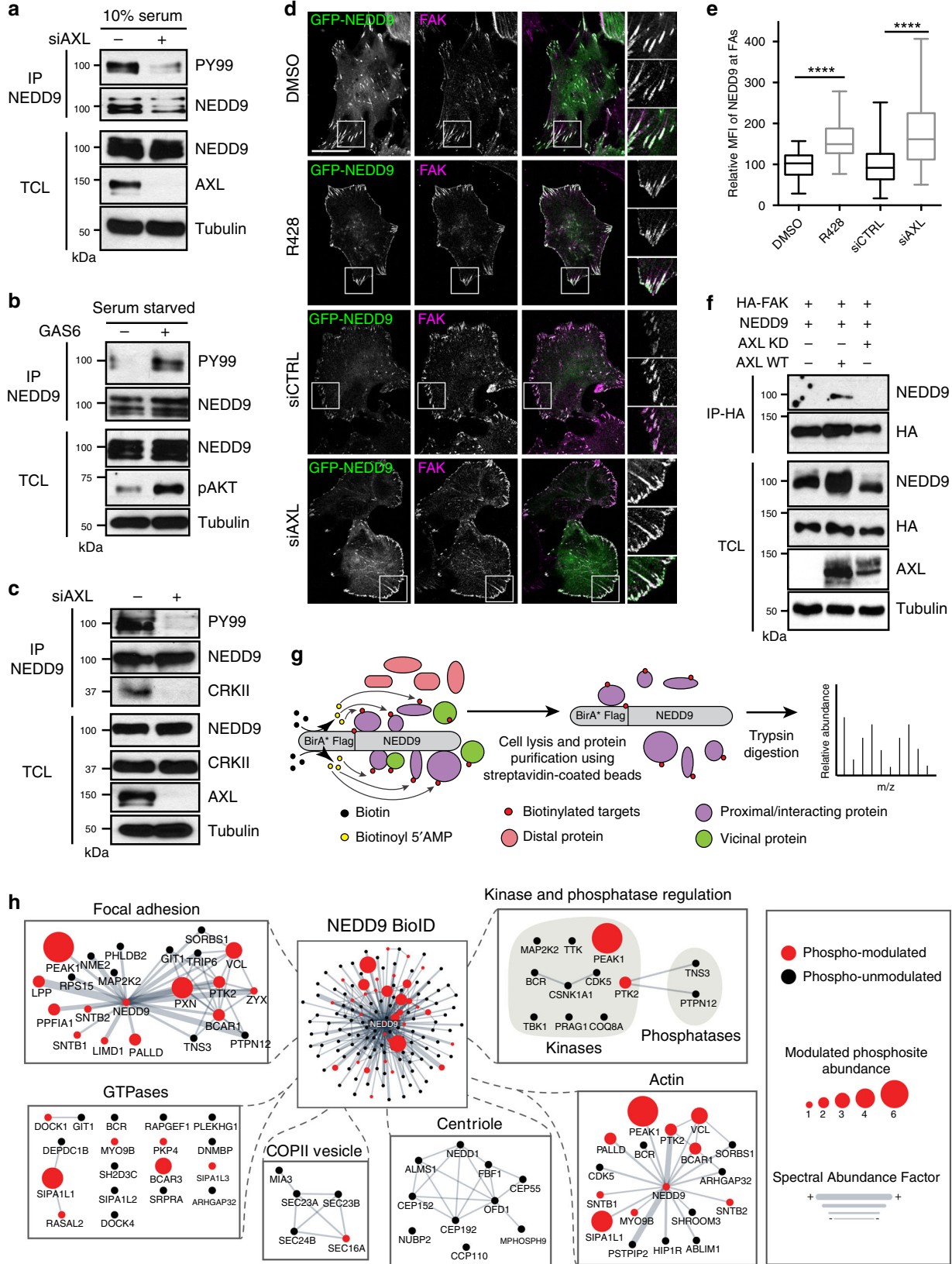

overexpression of PEAK1 (Fig. 7b). An in vitro kinase assay with recombinant proteins containing either the PXN N-terminus or C-terminus as substrates revealed the presence of a kinase activity in PEAK1 immunoprecipitates that is able to phosphorylate the C-terminus of PXN (Supplementary Fig. 5f, g). A recent study

showed that PRAG1 induces phosphorylation of cellular proteins by associating with the CSK tyrosine kinase[34,35]. We detected CSK in PEAK1 immunoprecipitates in an AXL-dependent manner in TNBCs (Fig. 7c) and found that CSK can promote phosphorylation of PXN (Fig. 7d). In contrast to PEAK1$^{WT}$, the

**Fig. 4 AXL phosphorylates NEDD9 and regulates its localization at FAs. a** AXL knockdown decreases NEDD9 phosphorylation levels. MDA-MB-231 cells were transfected with 100 nM siAXL. Following anti-NEDD9 immunoprecipitation, NEDD9 phosphorylation levels was determined by western blotting. **b** AXL activation increases NEDD9 phosphorylation levels. Serum-starved MDA-MB-231 cells were treated with GAS6 for 20 min. Following anti-NEDD9 immunoprecipitation, NEDD9 phosphorylation levels were determined by western blotting. **c** AXL knockdown decreases NEDD9/CRKII complex formation. MDA-MB-231 were transfected with 100 nM siAXL. Following anti-NEDD9 immunoprecipitation, levels of CRKII binding to NEDD9 was determined by western blotting. **d** NEDD9 localization at FAs is regulated by AXL. Representative confocal images of Hs578T cells transfected with GFP-NEDD9, plated on coverslips, and either treated with DMSO or 1 μM R428 or transfected with 100 nM siAXL. Cells were then fixed and permeabilized and stained for GFP (green) and FAK (magenta). Boxes are used to depict the location of the zoomed images. Scale bar, 20 μm. **e** Quantification of mean fluorescence intensity (MFI) of NEDD9 at FAK-positive FAs shown in **d**. Data are represented as boxplots where the middle line is the median, the lower and upper hinges correspond to the first and third quartiles, the upper whisker extends from the hinge to the largest value, and the lower whisker extends from the hinge to the smallest value. Two-tailed Mann–Whitney test was used. ****$P < 0.0001$ (15 cells per condition per experiment). Source data are provided as a Source data file. **f** AXL kinase activity regulates NEDD9 complex formation with FAK. Lysates of 293T cells expressing the indicated plasmids were subjected to anti-HA FAK immunoprecipitation. Co-immunoprecipitates were detected via western blotting. **g** Schematic workflow of the proximity-dependent biotinylation (BioID) proteomics approach performed with NEDD9 in Flp-In™ T-REx™ 293 cells. NEDD9 is fused to the promiscuous BirA* biotin ligase that can label the protein environment of the bait. **h** Network layout of the NEDD9 BioID dataset. Surrounding subnetworks in zoom boxes exhibit selected and relevant functions, such as "FA" and "Actin" where NEDD9 seems to play a central role. Node sizes represent the relative amount of GAS6-modulated phosphosites. Edge thickness represents the relative protein abundance depicted by the SAF (Spectral Abundance Factor) metric. The color of the node indicates if a NEDD9 prey has been phospho-modulated or not in the GAS6 phosphoproteomic screen (see Fig. 2).

---

PEAK1[3PA] mutant failed to promote PXN phosphorylation, confirming the necessity of PXN/CRKII complex formation to mediate PXN phosphorylation (Fig. 7e). To address the necessity of PEAK1 in AXL-induced FA turnover, we depleted *PEAK1* levels by siRNA in MDA-MB-231 cells expressing GFP-tagged PXN (GFP-PXN) and assessed the role of GAS6 on FA turnover by live cell imaging. We observed no decrease in FA disassembly time and lifetime in *PEAK1* knockdown in GFP-PXN expressing MDA-MB-231 cells after GAS6 stimulation (Fig. 7f–h; Supplementary Video 3), suggesting that AXL's regulation of FA turnover is mediated by PEAK1.

**AXL/PEAK1 recruit the FA turnover machinery to PXN FAs.** Since AXL activity led to the modulation of FA disassembly rate, we investigated whether this modulation is due to the recruitment of a disassembly complex machinery to FA. It has been shown that FA turnover is mediated in part by the recruitment of the βPIX/GIT1/PAK1 complex into PXN FAs to induce disassembly of the FA structure[36,37]. Interestingly, our phospho-screen data revealed βPIX to be modulated by GAS6-induced AXL activation on S703, which is located in its RhoGEF domain (Supplementary Data 1). In addition, we found GAS6-mediated activation of AXL promotes the activation of PAK kinases while its inhibition by R428 decreases it (Supplementary Fig. 6a), suggesting that AXL modulation of βPIX/GIT1/PAK1 complex phosphorylation levels regulates their recruitment and activity at FA sites. To determine whether AXL modulates FA turnover by regulating the recruitment of this complex to FAs, we stained MDA-MB-231 or Hs578T cells, treated with R428, GAS6, or transfected with siAXL, for either GIT1 or βPIX, and assessed their localization at PXN FAs. Even though the number of PXN FAs is higher, a significant decrease in βPIX/GIT1 recruitment to PXN FAs upon R428 treatment or knockdown of *AXL* was observed, whereas GAS6 treatment increased the recruitment of this complex to the FA sites (Fig. 8a, b; Supplementary Fig. 6b, c). This suggests that AXL activity induces FA turnover by modulating the recruitment of the βPIX/GIT1/PAK1 complex to FA sites. Similarly, knockdown of *βPIX/GIT1/PAK1* and *CDC42*, a known target of βPIX and an activator of PAK kinases[38,39], in TNBC cells led to an increase in FA number and mechanistically to a slower disassembly, similar to what was observed previously with AXL inhibition (Supplementary Fig. 6d, e). These results reinforce the connection between AXL signaling, the βPIX/GIT1/PAK1 complex, and the promotion of FA disassembly.

We hypothesized that CRKII-induced FA turnover could be mediated by PEAK1 functioning as a scaffold to coordinate the recruitment of the βPIX/GIT1/PAK1 complex to phospho-PXN downstream of AXL. Indeed, complex formation between PEAK1 and both βPIX and GIT1 was detectable (Fig. 8c, Supplementary Fig. 6f), and modulating *PEAK1* expression levels via siRNA in MDA-MB-231 cells revealed a decrease in GIT1 recruitment (Fig. 8d–g) and βPIX localization to PXN FAs (Supplementary Fig. 6g, h). These data support a model where AXL signaling promotes the recruitment of the βPIX/GIT1/PAK1 into PXN FAs through the CRKII/PEAK1 module.

**AXL/PEAK1 signaling promotes metastasis in vivo.** To assess the function of PEAK1 in an in vivo context, we used CRISPR-CAS9 to delete *PEAK1* in the MDA-MB-231 luciferase-expressing cell model (PEAK1[KO]) (Fig. 9a). Bioluminescence assays revealed that PEAK1[KO] cells had a decreased ability to migrate and invade in comparison to control cells (Supplementary Fig. 7a, b). To investigate whether PEAK1 plays a role in tumor growth and metastasis in vivo, PEAK1[KO] cells, generated from two different single guide RNAs (sgRNAs), as well as parental cells were injected into mammary fat pads of nude mice and tumor growth was followed for 4 weeks (Fig. 9b). PEAK1[KO] cells showed delayed tumor growth, which was recapitulated in vitro in two-dimensional MTT proliferation and tumorsphere assays (Fig. 9c; Supplementary Fig. 7c–f). Mice bearing PEAK1[KO] tumors showed no lung metastases when compared with mice harboring WT tumors (Fig. 9d). The number of circulating tumor cells (CTCs) was lower in mice bearing PEAK1[KO] tumors in comparison to mice harboring WT tumors (Fig. 9e). To bypass the primary tumor growth defect, we conducted an in vivo experimental metastasis assay by injecting MDA-MB-231-Luc WT or PEAK1[KO] cells into the lateral tail vein of nude mice. Metastasis progression was followed for 7 weeks by bioluminescence imaging. Knockout of PEAK1 in MDA-MB-231-Luc cells significantly reduced the ability of the cells to colonize lungs (Fig. 9f, g). Collectively, these data suggest that PEAK1 is required for both tumor growth and metastasis in a TNBC cellular model.

To determine whether the AXL/PEAK1 connection is required for tumor growth and metastasis, we used the PEAK1[3PA] mutant that we demonstrated to be uncoupled from AXL-induced phosphorylation (Fig. 6a–c). We rescued the PEAK1[KO] cells with either empty vector (EV), WT PEAK1 (PEAK1[WT]), or mutant PEAK1 (PEAK1[3PA]) (Fig. 10a). PEAK1 re-expression in PEAK1[KO] cells was able to significantly increase migration,

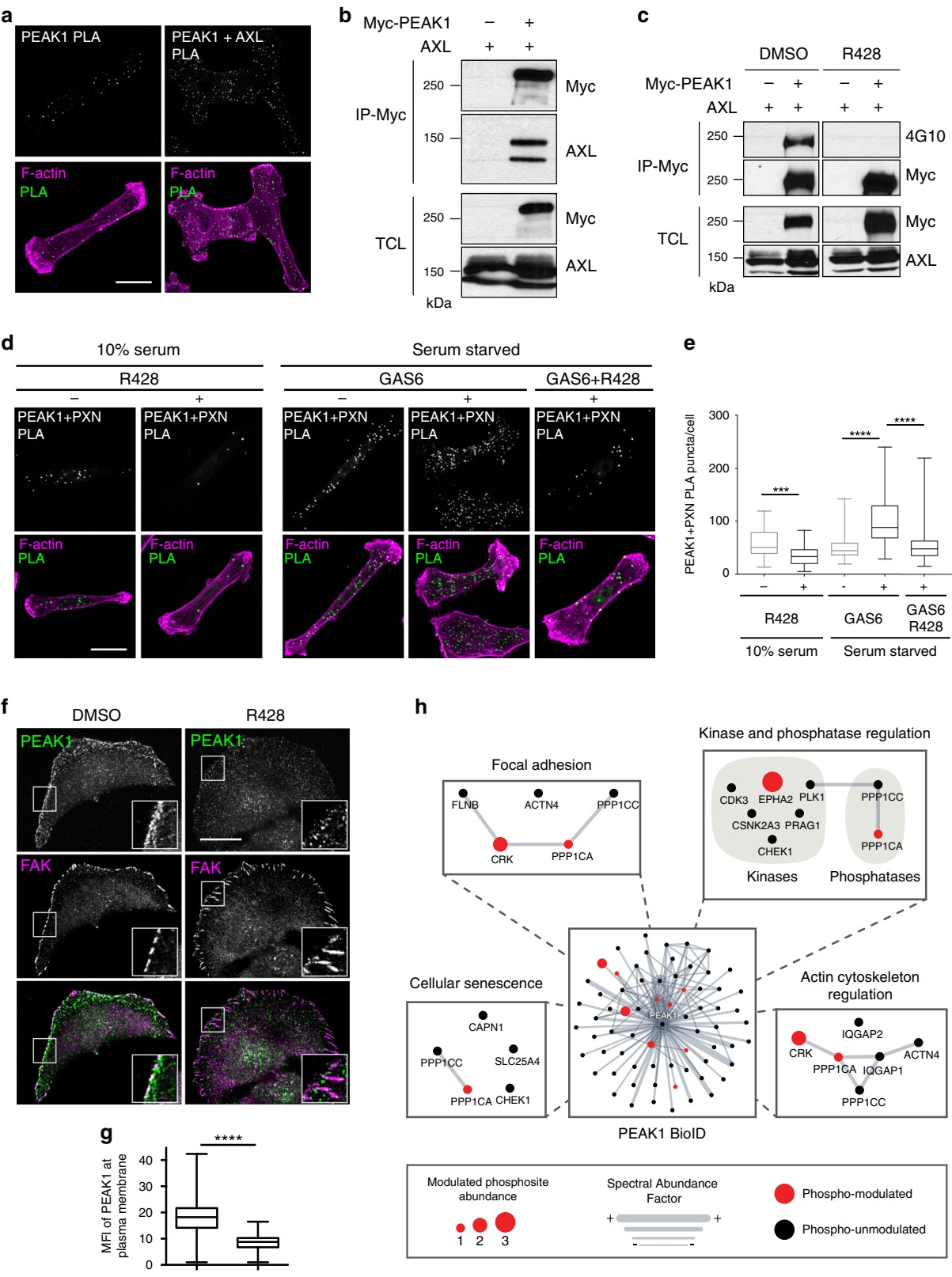

invasion, and wound healing capabilities of PEAK1$^{KO}$ cells (Supplementary Fig. 7g–i). However, the expression of PEAK1$^{WT}$ rescued cell migration and invasion of PEAK1$^{KO}$ cells more significantly than the expression of PEAK1$^{3PA}$. To investigate whether the PEAK1$^{3PA}$ mutant can salvage tumor growth and metastasis in vivo, PEAK1$^{KO}$ rescued cells were injected into mammary fat pads of nude mice and tumor growth was followed for 8 weeks (Fig. 10b). Tumor growth was rescued equally with PEAK1$^{WT}$ and PEAK1$^{3PA}$ compared to PEAK1$^{KO}$, where similar tumor volumes were obtained (Fig. 10c, Supplementary Fig. 7j). Interestingly, mice bearing tumors expressing PEAK1$^{WT}$ showed more spontaneous lung metastases when

**Fig. 5 AXL interacts with PEAK1 and modulates its phosphorylation and localization. a** PEAK1 localizes with AXL in TNBC cells. Representative confocal images of MDA-MB-231 cells. PLA was used to analyze the proximity localization of PEAK1 and AXL. Scale bar, 20 μm. **b** PEAK1 interacts with AXL. Lysates of 293T cells expressing the indicated plasmids were used for anti-Myc immunoprecipitation. Levels of AXL is detected via western blotting. **c** PEAK1 is phosphorylated by AXL. Lysates of 293T cells expressing the indicated plasmids and treated or not with 1 μM R428 for 1 h were subjected to anti-Myc immunoprecipitation. Levels of Myc-PEAK1 tyrosine phosphorylation was detected via western blotting. **d** PEAK1 is localized at PXN FA sites. Representative confocal images of MDA-MB-231 cells treated with DMSO or 1 μM R428, GAS6, or GAS6 and R428. PLA was used to analyze proximity localization of PEAK1 at PXN-positive FAs. Scale bar, 20 μm. **e** Quantifications of the number of PLA puncta per cell per condition. \*\*\*$P = 0.000014$; \*\*\*\*$P < 0.000001$ ($n = 3$ experiments, 15 cells per condition per experiment). **f** PEAK1 recruitment to the cell membrane is AXL regulated. Representative confocal images of Hs578T cells treated with DMSO or 1 μM R428. Cells were stained for PEAK1 (green) and FAK (magenta). Boxes are used to depict the location of the zoomed images. Scale bar, 20 μm. **g** Quantification of mean fluorescence intensity (MFI) of PEAK1 at periphery of membrane shown in **f**. \*\*\*\*$P < 0.0000001$ (15 cells per condition per experiment). **h** Network layout of the PEAK1 BioID dataset. Surrounding subnetworks in zoom boxes exhibit selected and relevant functions, such as "FA" and "Actin" where PEAK1 seems to play a role. Node sizes represent the relative amount of GAS6-modulated phosphosites. Edge thickness represents the relative protein abundance depicted by the SAF (Spectral Abundance Factor) metric. The color of the node indicates if a PEAK1 prey has been phospho-modulated or not in the GAS6 phosphoproteomic screen (see Fig. 2). Data are represented as boxplots where the middle line is the median, the lower and upper hinges correspond to the first and third quartiles, the upper whisker extends from the hinge to the largest value, and the lower whisker extends from the hinge to the smallest value (**e**, **g**). Two-tailed Mann–Whitney test was used to calculate $P$ value. Source data are provided as a Source data file.

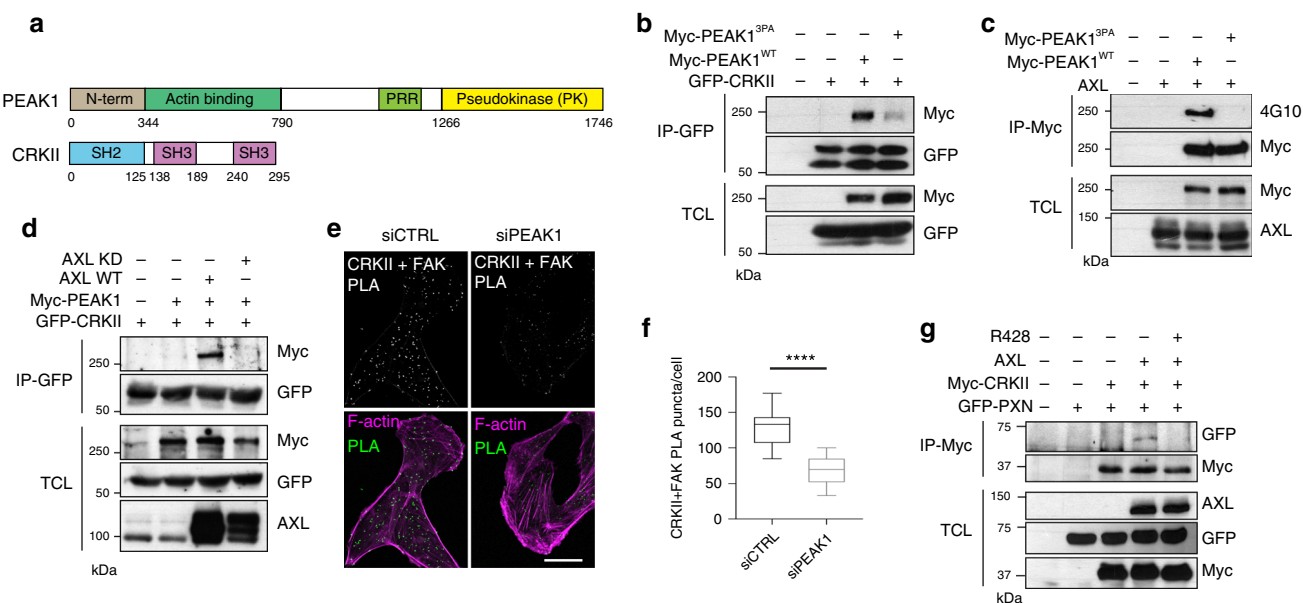

**Fig. 6 CRKII/PEAK1 binding mediates PEAK1 phosphorylation by AXL and CRKII localization at FAs. a** Schematic of PEAK1 and CRKII domains. **b** CRKII binds PEAK1 proline-rich region (PRR). Lysates of 293T cells expressing the indicated plasmids were used for anti-GFP immunoprecipitation. Levels of Myc-PEAK1 is detected via western blotting. **c** Lysates of 293T cells transfected with the indicated plasmids were subjected to anti-Myc immunoprecipitation. Myc-PEAK1 phosphorylation levels were detected via western blotting. **d** Lysates of 293T cells transfected with the indicated plasmids were subjected to anti-GFP immunoprecipitation. Levels of Myc-PEAK1 were detected via western blotting. **e** PEAK1 regulates CRKII localization at FA sites. Representative confocal images of MDA-MB-231 cells transfected with either 100 nM siCTRL or siPEAK1. PLA was used to analyze proximity localization of CRKII at FAK-positive FAs. Scale bar, 20 μm. **f** Quantifications of the number of PLA puncta per cell per condition. \*\*\*\*$P < 0.000001$ ($n = 3$ experiments, 15 cells per condition per experiment). Data are represented as boxplots where the middle line is the median, the lower and upper hinges correspond to the first and third quartiles, the upper whisker extends from the hinge to the largest value, and the lower whisker extends from the hinge to the smallest value. Two-tailed Mann–Whitney test was used to calculate $P$ value. Source data are provided as a Source data file. **g** AXL modulates CRKII binding to PXN. Lysates of 293T cells expressing the indicated plasmids were used for anti-Myc immunoprecipitation. Levels of GFP-PXN is detected via western blotting.

compared with mice harboring PEAK1$^{KO}$ tumors expressing either EV or PEAK1$^{3PA}$ (Fig. 10d, Supplementary Fig. 7k). The number of CTCs was significantly higher in mice bearing PEAK1$^{WT}$ tumors in comparison to mice harboring PEAK1$^{KO}$ tumors expressing EV (Fig. 10e). In addition, mice injected with PEAK1$^{3PA}$ cells had a significant decrease in the number of CTCs when compared with those injected with PEAK1$^{WT}$ cells although both had a similar tumor size, suggesting a defect in the intravasation of the PEAK1$^{3PA}$ tumor cells. We further assessed whether the enhanced metastasis seen in PEAK1$^{WT}$ tumors was due to a difference in their FA dynamics by analyzing their PXN-containing FAs. Indeed, PEAK1$^{KO}$ cells had a

significant increase in the number and size of FAs compared to parental MDA-MB-231 cells (Fig. 10f–h). Interestingly, rescue with expression of PEAK1$^{WT}$ dramatically decreased the number and size of FAs, and although the expression of PEAK1$^{3PA}$ did result in a smaller number of adhesions compared to PEAK1$^{KO}$ cells, these appeared larger. One possible explanation is that the large adhesions observed in the PEAK1$^{3PA}$ cells are due to a defect in FA disassembly that is not present in the PEAK1$^{WT}$ cells. Indeed, PEAK1 binding to GIT1 or βPIX was decreased upon the expression of PEAK1$^{3PA}$ as assessed by co-IP (Supplementary Fig. 7l, m). Collectively, these results demonstrate that the AXL/PEAK1 signaling complex is essential for the

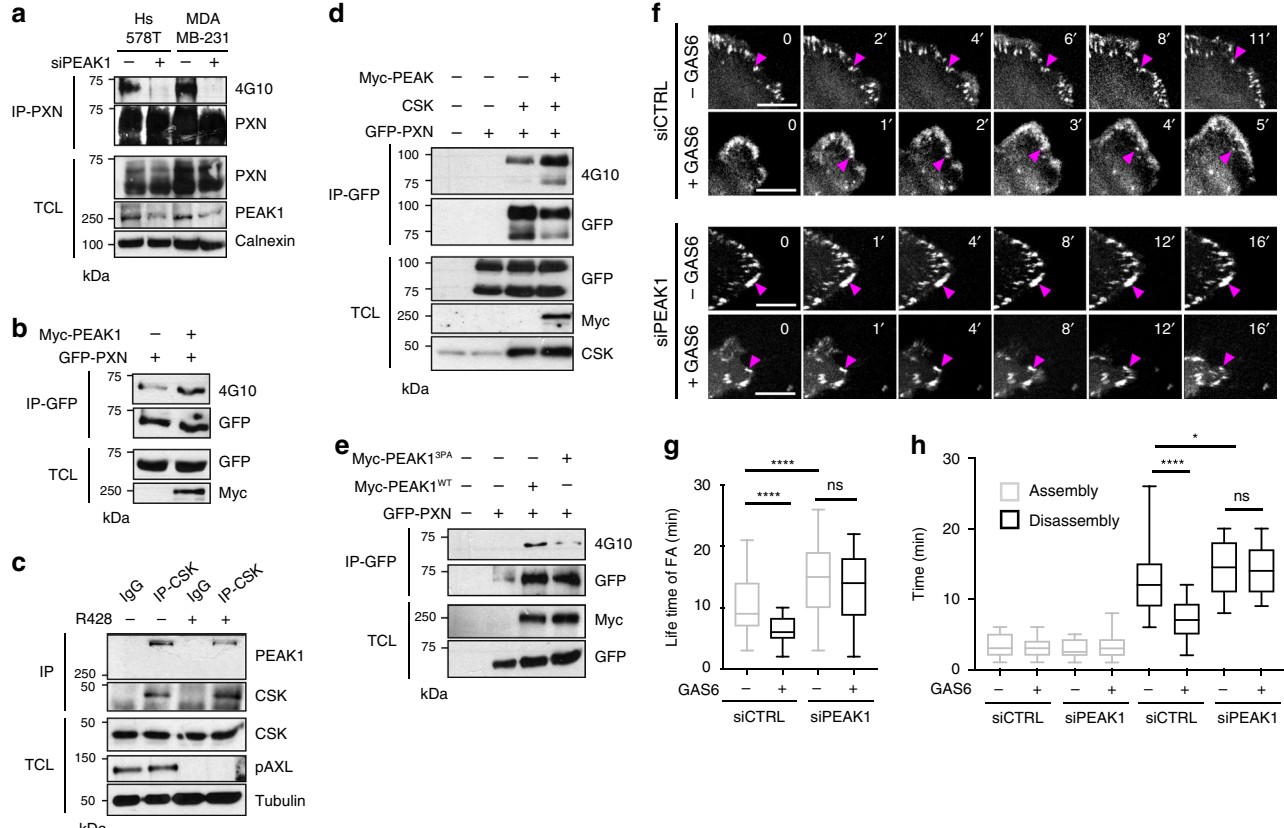

**Fig. 7 PEAK1 regulates FA turnover by PXN phosphorylation downstream of AXL. a** PXN phosphorylation is PEAK1 dependent in TNBC cells. Hs578T and MDA-MB-231 cells were transfected with 100 nM siCTRL or siPEAK1 and their lysates were used for anti-PXN immunoprecipitation. Phosphorylation levels of PXN was analyzed via western blotting. **b** PEAK1 expression increases PXN phosphorylation. Lysates of 293T cells expressing the indicated plasmids were used for anti-GFP immunoprecipitation. Levels of PXN phosphorylation is detected via western blotting. **c** PEAK1 interacts with CSK in an AXL-dependent manner. Lysates of MDA-MB-231 cells treated or not with 1 μM R428 were used for anti-CSK immunoprecipitation. PEAK1 levels in the IP were analyzed via western blotting. **d** Lysates of 293T cells transfected with the indicated plasmids were subjected to anti-GFP immunoprecipitation. GFP-PXN phosphorylation levels were detected via western blotting. **e** Lysates of 293T cells transfected with the indicated plasmids were subjected to anti-GFP immunoprecipitation. GFP-PXN phosphorylation levels were detected via western blotting. **f** MDA-MB-231 GFP-PXN-expressing cells transfected with 100 nM of siCTRL or siPEAK1 and treated with or without GAS6 were imaged live by spinning disk microscopy for a period of 30 min to assess the dynamics of FA turnover. Magenta arrow heads point toward an FA that was followed for its assembly and disassembly times. Scale bar, 5 μm. **g, h** AXL modulation of FA turnover is PEAK1 mediated. Quantification of assembly and disassembly time (**g**) and lifetime (**h**) of the FAs of GFP-PXN expressing MDA-MB-231 cells. ****$P < 0.000001$; *$P = 0.0196$ ($n = 3$ experiments with 90 FAs followed per condition). Data are represented as boxplots where the middle line is the median, the lower and upper hinges correspond to the first and third quartiles, the upper whisker extends from the hinge to the largest value, and the lower whisker extends from the hinge to the smallest value. Two-tailed Mann–Whitney test was used to calculate $P$ value. Source data are provided as a Source data file.

cell migration/invasion and FA turnover that lead to metastasis, yet this complex is not required for tumor growth.

## Discussion

The current understanding of AXL signaling involves the activation of downstream signaling intermediates shared with other RTKs[40] such as AKT, but fails to explain the unique pro-invasion influence of AXL. By performing an AXL phosphoproteomic screen, we explain how this RTK facilitates cancer cell invasion and metastasis by strongly modulating the biological process of FA turnover (Fig. 10i). While we report that AXL can be detected in proximity to FA proteins, further work is required to determine whether AXL traffics into these structures or whether it phosphorylates its targets prior to their localization into FAs. At a molecular level, AXL activation leads to a modulation of the FA complex NEDD9/CRKII and their partner PEAK1. We propose that CRKII recruits PEAK1 into FAs, although it is also plausible that it may stabilize its localization there. We discovered that PEAK1 at PXN FA sites is a major mediator of PXN tyrosine

phosphorylation, via its formation of a complex with the kinase CSK. We show that PEAK1 is phosphorylated on at least five tyrosine residues upon AXL activation, and it will be important in the future to map how these events contribute to FA turnover. The FA disassembly complex βPIX/GIT1/PAK1 was demonstrated to act downstream of AXL and PEAK1 to ultimately increases FA turnover. PRAG1 was recently found to engage in a phosphotyrosine-dependent interaction with CSK and to promote CSK kinase activity, yet no specific substrate was identified[34]. We suspect that PEAK1 heterodimerizes with PRAG1, to interact with CSK. Still, our studies reveal a specific protein, PXN, being phospho-modulated on tyrosine residues by the PEAK1 pseudo-kinase. Interestingly, the receptor tyrosine kinase EPHA2 was found to be phosphorylated by AXL from our phosphoproteomic data and our BioID data demonstrated that it interacts with WT, but not mutant, PEAK1. A recent study has shown EPHA2 signaling promotes cell motility by modulating FA dynamics[41]. This raises the possibility that AXL could heterodimerize and transactivate EPHA2 that may contribute to PEAK1

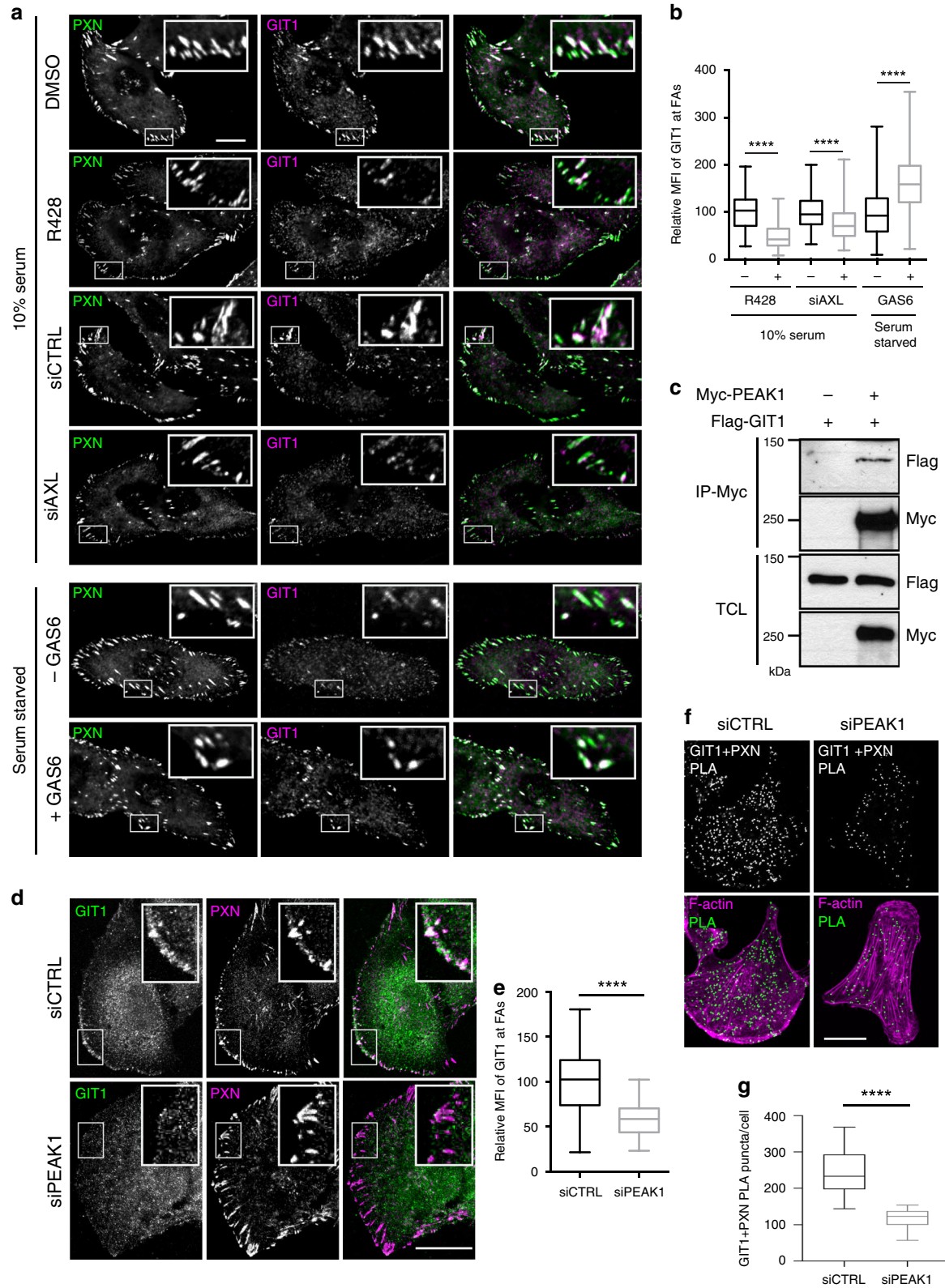

phosphorylation and modulate FA turnover. We also found that the PEAK1/CRKII interaction plays a significant role not only in FA dynamics but also in TNBC metastasis. PEAK1 binding to CRKII was found to be essential for AXL-induced PEAK1 phosphorylation and uncoupling of AXL/PEAK1 impaired cell migration and invasion in vitro and metastasis in vivo. Yet, the connection of AXL to PEAK1 was completely dispensable for

rescuing tumor growth, demonstrating the importance of AXL/ PEAK1 specifically in the invasion process. Like AXL, PEAK1 expression correlates in many cancer models with mesenchymal features, disease relapse, and therapy resistance[42,43]. Developing strategies to interfere with pseudo-kinases, such as PEAK1, by pharmacologically targeting their bound kinases may prove useful to block tumor growth and metastasis. AXL is not expressed in all

**Fig. 8 AXL/PEAK1 complex regulates recruitment of FA disassembly complex. a** Representative confocal images of MDA-MB-231 cells treated with DMSO or 1 µM R428, transfected with 100 nM of siCTRL or siAXL, or serum starved and treated with GAS6 for 20 min. Cells were stained for PXN (green) and GIT1 (magenta). Boxes are used to depict the location of the zoomed images. Scale bar, 10 µm. **b** Quantification of mean fluorescence intensity (MFI) of GIT1 at PXN-positive FAs shown in **a**. ****P < 0.000001 (15 cells per condition per experiment). **c** PEAK1 interacts with GIT1. Lysates of 293T cells expressing the indicated plasmids were used for anti-Myc immunoprecipitation. Levels of Flag-GIT1 is detected via western blotting. **d** PEAK1 regulates GIT1 localization at FA sites. Representative confocal images of MDA-MB-231 cells transfected with either 100 nM siCTRL or siPEAK1. Cells were stained for GIT1 (green) and PXN (magenta). Scale bar, 20 µm. **e** Quantification of mean fluorescence intensity (MFI) of GIT1 at PXN-FAs shown in **d**. ****P < 0.000001 (15 cells per condition per experiment). **f** PEAK1 expression levels modulate GIT1 recruitment to FA sites. Representative confocal images of MDA-MB-231 cells transfected with 100 nM siCTRL or siPEAK1. PLA was used to analyze the proximity localization of GIT1 at PXN-FAs. Scale bar, 20 µm. **g** Quantifications of the number of PLA puncta per cell per condition. ****P < 0.000001 (n = 3 experiments, 15 cells per condition per experiment). Data are represented as boxplots where the middle line is the median, the lower and upper hinges correspond to the first and third quartiles, the upper whisker extends from the hinge to the largest value, and the lower whisker extends from the hinge to the smallest value (**b**, **e**, **g**). Two-tailed Mann–Whitney test was used to calculate P value. Source data are provided as a Source data file.

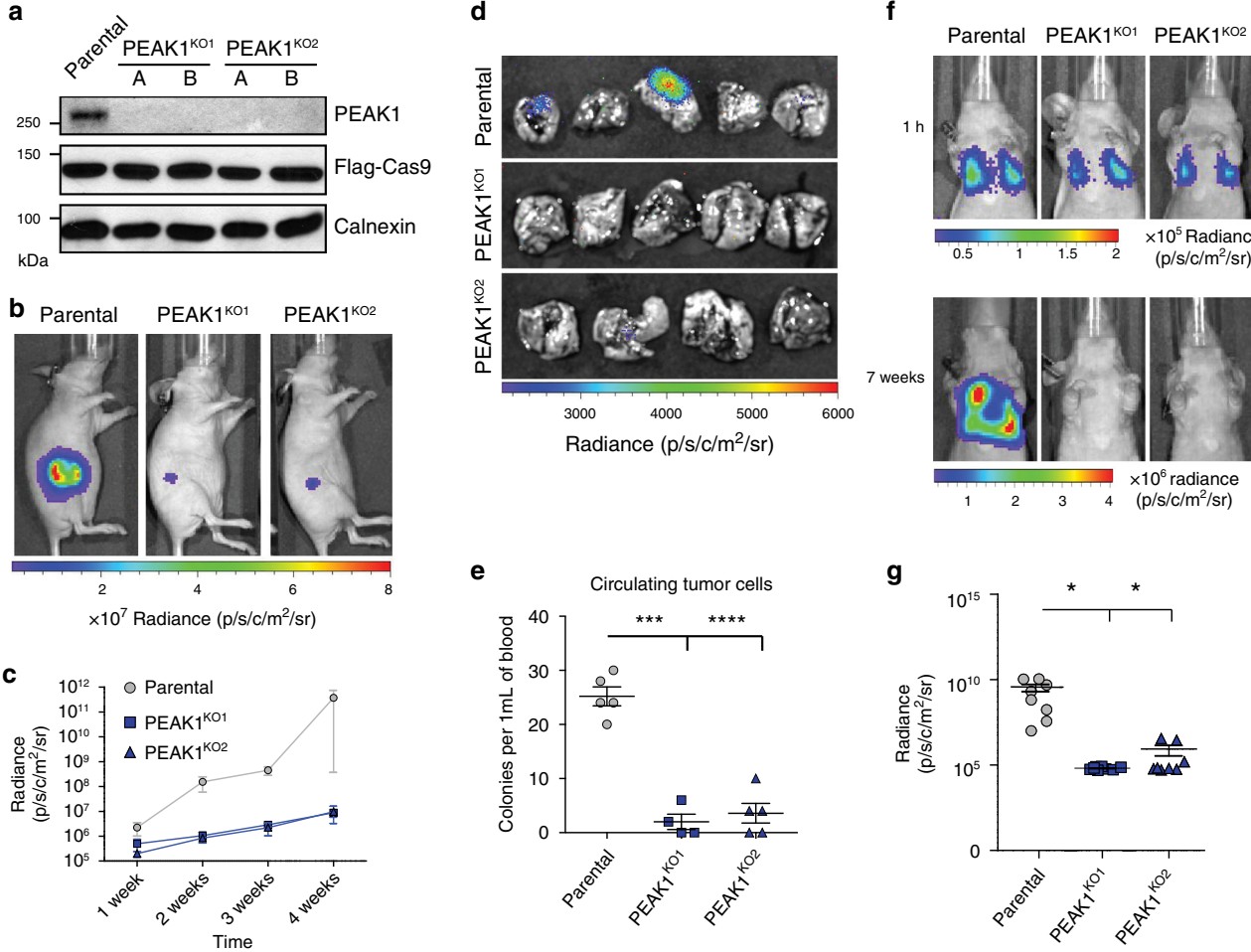

**Fig. 9 PEAK1 expression regulates tumor growth and metastasis of TNBC in vivo. a** Lysates of MDA-MB-231-Luc CRISPR PEAK1 knockout clones. **b** Representative in vivo bioluminescent images of mammary fat pad-injected mice with MDA-MB-231-Luc WT (parental) or CRISPR PEAK1 KO cells 4 weeks postinjection. **c** Bioluminescence quantification of tumor growth at different weeks postinjection of mice bearing parental (n = 5), PEAK1 KO1 (n = 5), or PEAK1 KO2 (n = 5). P < 0.0001. **d** Representative bioluminescent lung images of mice shown in **b**. For the lungs of the mice bearing CRISPR PEAK1 KO tumors, lungs were dissected once the tumor reached the size of the WT tumors. **e** Circulating tumor cells isolated from mice-bearing parental or PEAK1 KO mammary tumors. ***P = 0.000022; ****P = 0.000027 (n = 5 for each group). **f** PEAK1 regulates metastasis of TNBC cells in vivo. Representative in vivo bioluminescent images of tail vein-injected mice with parental or CRISPR PEAK1 KO cells 1 h and 7 weeks postinjection. **g** Bioluminescence quantification of lung metastases 7 weeks postinjection of mice bearing parental (n = 8), PEAK1 KO1 (n = 8), or PEAK1 KO2 (n = 8). *P = 0.0439. Data were analyzed from 5 mice (**c**, **e**) or 8 mice (**g**) for each group and expressed as mean ± s.e.m. Two-tailed unpaired t test was used (**c**, **e**, **g**). Source data are provided as a Source data file.

primary breast cancers analyzed[10]. Its expression in HER2 breast cancer is linked to EMT[11]. Recently, it was found that AXL and PEAK1 are co-expressed in aggressive basal B breast cancers where AXL is required for invasion[44]. Hence, future work should aim to determine the subtypes of breast cancer that would be candidates to AXL or PEAK1 inhibition to limit metastasis.

The effects of AXL activation are not limited to FA dynamics. We have also discovered links to regulation of the actin

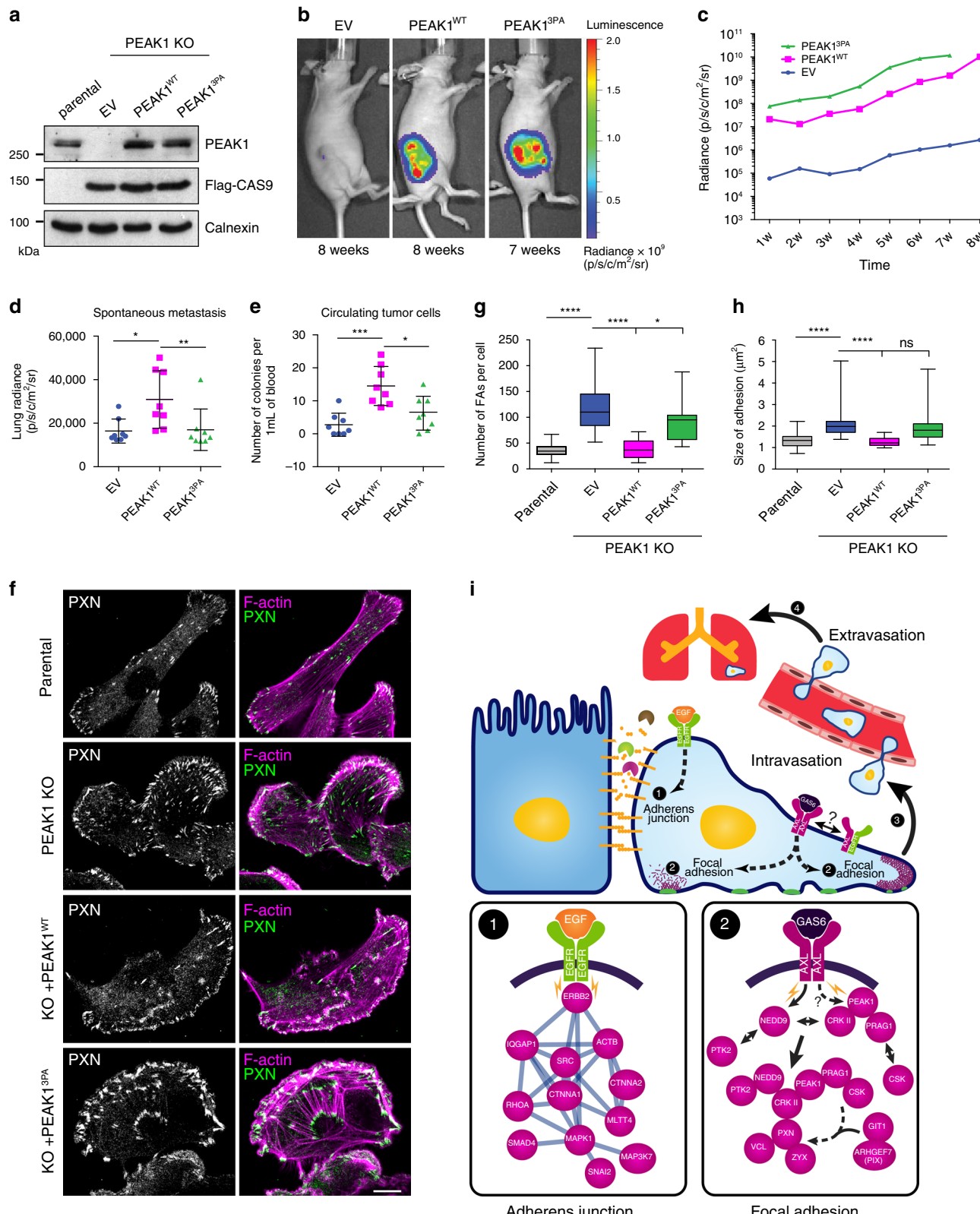

cytoskeleton, GTPase regulation, and phagocytosis. It will be important to compare the activation of these other networks by AXL and other RTKs and to probe their function in cell invasion. In addition, we used soluble GAS6 as a tool for AXL activation. The experimental model that we chose here, namely, activation of AXL by its ligand GAS6, has the advantage of being controllable in vitro, yet it is most likely a simplified model of the diversity of interactions that AXL can engage in at the cell surface to become activated. For instance, a pool of AXL is reported to co-signal with EGFR in TNBC cells to diversify signaling[45,46]. Overall, defining the phosphoproteome of EGFR/AXL in the future may reveal additional therapeutic targets to limit metastasis.

**Fig. 10 Uncoupling PEAK1 from AXL signaling leads to impaired metastasis. a** Lysates of rescued MDA-MB-231-Luc CRISPR PEAK1 knockout cells. **b** Representative in vivo bioluminescent images of fat pad-injected mice with MDA-MB-231-Luc PEAK1 knockouts rescued with EV, PEAK1 WT, or PEAK1 3PA mutant 8 and 7 weeks postinjection, respectively. **c** Bioluminescence quantification of tumor growth at different weeks postinjection of mice bearing PEAK1 KO rescued with EV ($n = 8$), PEAK1 WT ($n = 8$), or PEAK1 3PA ($n = 8$). **d** Bioluminescence quantification of the spontaneous metastases in the lungs measured ex vivo of the fat pad-injected mice shown in **c** when the tumors from PEAK1$^{WT}$ and PEAK1$^{3PA}$ reached the same size (7–8 weeks postinjection). *$P = 0.0299$; **$P = 0.0125$. $n = 8$ for EV, $n = 8$ for PEAK1 WT, and $n = 8$ for PEAK1 3PA. **e** Circulating tumor cells isolated from mice-bearing EV PEAK1$^{KO}$, PEAK1$^{WT}$, or PEAK1$^{3PA}$ rescued mammary tumors. *$P = 0.0098$, ***$P = 0.0003$ ($n = 8$ for each group). **f** Representative confocal images of parental MDA-MB-231 cells and PEAK1 KO cells rescued with EV, PEAK1 WT, or PEAK1 3PA. Cells were stained for PXN (green) and Phalloidin (magenta). Scale bar, 10 μm. **g** Quantification of the FA number per cell depicted in **f**. *$P = 0.014$; ****$P < 0.000001$ ($n = 3$ experiments with 30 cells per condition). **h** Quantification of the size of the adhesions quantified in **g**. ****$P < 0.000001$ ($n = 3$ experiments with 30 cells per condition). Data are represented as boxplots where the middle line is the median, the lower and upper hinges correspond to the first and third quartiles, the upper whisker extends from the hinge to the largest value, and the lower whisker extends from the hinge to the smallest value (**g**, **h**). Two-tailed Mann–Whitney test was used to calculate $P$ value. Source data are provided as a Source data file. **i** Schematic model of AXL and EGFR signaling in a cancer cell, where EGFR modulates adherens junctions and AXL modulates the FA turnover. Upon EGF stimulation, EGFR induces the expression of MMPs to degrade the adherens junctions. Once cell–cell contact is lost, AXL activation by GAS6 or EGFR can lead to the modulation of FAs turnover at the rear or front end of the cell. In specific, AXL directly phosphorylates NEDD9 to recruit its complex formation with PTK2 (FAK) and CRKII. Simultaneously, AXL can phosphorylate PEAK1, in complex with PRAG1 and CRKII, and recruit PTK2/NEDD9/CRKII/PEAK1/PRAG1 to PXN-positive FAs. Upon PEAK1 phosphorylation, CSK is recruited to PEAK1/PRAG1 complex at FAs to induce tyrosine phosphorylation of PXN. In addition, PEAK1 can also recruit βPIX/GIT1 complex to PXN-positive FAs to induce FA disassembly and turnover.

## Methods

**Cell culture, treatments, transfections, and infections**. MDA-MB-231, Hs578T, HEK293T (293T), and HeLa cells were cultured in Dulbecco's modified Eagle's medium (DMEM), supplemented with 10% fetal bovine serum (FBS; Gibco) and 1% penicillin/streptomycin antibiotics (Wisent). Flp-In T-REx™ 293 cells were obtained from Life Technologies. MDA-MB-231, Hs578T, and HeLa cells were obtained from ATCC and were transfected using Lipofectamine 2000 (Invitrogen) according to the manufacturer's instructions. 293T cells were transfected using calcium phosphate precipitation. MDA-MB-231 and Hs578T cells were serum starved overnight prior to treatment with GAS6 recombinant media for the indicated time. These cells were also treated with 1 μM R428 (Apexbio) for 1 h. siRNA transfections were performed at a concentration of 100 μM for 72 h using the siRNA indicated in Supplementary Table 2. Lentiviral and retroviral infections were carried out by transfecting 293T cells using the plasmids indicated in Supplementary Table 3. Viral supernatants were harvested 24 h later and added to MDA-MB-231 cells in the presence of 10 μg/mL of polybrene.

**Plasmids**. The plasmid pcDNA-hGAS6-His, a kind gift from Dr. Ed Manser (A*STAR, Singapore), was used to generate pcDNA5-pDEST-hGAS6-His by Gateway cloning system using the primers indicated in Supplementary Table 4 to produce the GAS6 conditioned media. pEGFP-NEDD9 and pcDNA3.1-HA-NEDD9 were a kind gift from Dr. Erica Golemis (Fox Chase Cancer Center) and were used to generate pcDNA5-pDEST-BirA*-Flag-N-ter NEDD9 by Gateway cloning system using the primers indicated in Supplementary Table 4. pEGFP-NEDD9 was also used to generate pGEX4T1-hNEDD9 SD and pGEX4T1-hNEDD9 CT using the primers indicated in Supplementary Table 4 by Gibson Assembly cloning system. pOG44 was from Invitrogen. pcDNA5-pcDEST-BirA-Flag-N-ter were generated by the laboratory of Anne-Claude Gingras. pCIS2-HA-DOCK3 was a kind gift from Dr. David Schubert (Salk Institute). pCMV-sport6-mouse AXL-WT was a kind gift from Dr. Rob Screaton (Sunnybrook Institute, Toronto). pRetroX-Sgk269 WT[27] was used to generate pcDNA5-pDEST-BirA*-Flag-N-ter PEAK1 WT, pcDNA5-pDEST-BirA*-Flag-N-ter PEAK1 3PA, and pCS-6Myc-hPEAK1 using Gateway cloning system. pCS-6Myc-hPEAK1 3PA and pCS-6Myc-hPEAK1 Y531F were generated by site-directed mutagenesis using the primers indicated in Supplementary Table 4. pRetroX-Sgk269 WT[27] was used to generate pRetroX-Sgk269 WT KO1 PAM mutant and pRetroX-Sgk269 3PA KO1 PAM mutant by site-directed mutagenesis using the primers indicated in Supplementary Table 4. pCMV5-HA- ARHGEF7 was a kind gift from Dr. Liliana Attisano (University of Toronto, Toronto). pGEX2TK GST-PXN 1-313 and pGEX2TK GST-PXN 329-559 used in PXN in vitro kinase assay were generated as previously described[47]. pXM139-CSK was generated as previously described[48].

**Generation and quantification of GAS6 conditioned media using enzyme-linked immunosorbent assay (ELISA)**. Flp-In™ T-REx™ HeLa cells transfected with pOG44 and pcDNA5-pDEST-hGAS6-His were used to produced recombinant human GAS6 in a tetracycline-inducible manner. Cells were treated for 24 h with tetracycline and vitamin K$_3$ (4 μM, Sigma) or warfarin (10 μM, Santa Cruz Biotechnology Inc.) in DMEM without serum. The supernatant is collected and centrifuged to eliminate cells. GAS6 conditioned media was quantified by ELISA (DuoSet ELISA, R&D Systems). The γ-carboxylation state of GAS6 was assessed by western blot using mouse monoclonal antibody against human gamma-carboxyglutamyl (Gla) residues (Sekisui Diagnostics, Ref. 3570). We used anti-Histidine antibody to assess the production of recombinant GAS6 in each

condition. A 96-well plate was coated with 400 ng/mL of mouse anti-human GAS6 antibody and incubated overnight at room temperature (RT). Then the plate was washed three times with 0.05% Tween in phosphate-buffered saline (PBS; Wash Buffer) and blocked with 1% bovine serum albumin (BSA; Reagent Buffer) in PBS for 1 h at RT. Three additional washes were performed, and 2 μL of samples or standards (recombinant human GAS6; R&D Systems) diluted in Reagent buffer was loaded for 2 h in RT. Washing step was repeated with Wash Buffer, followed by the addition of 100 ng/mL of biotinylated goat anti-human GAS6 antibody for 2 h at RT. Detection was performed using streptavidin conjugated to horseradish peroxidase for 20 min. After the last washing step, tetramethylbenzidine was incubated for 15 min. The reaction was stopped with HCl 1 M, and the optical density of each well was determined using a microplate reader set at 450 nm. Concentration of GAS6 in the samples were calculated from polynomial second-order standard curve obtained from the standard included in ELISA assay.

**Preparation of liposomes**. Lipids were purchased from Sigma (catalog #790304P) and were solubilized for cell delivery as previously published[49]. Briefly, we resuspended a dried lipid film in DMEM media with 0.35% BSA, vortexed vigorously, and then diluted it to the indicated concentration (Liposome #1—10 μg/mL and Liposome #2—100 μg/mL). In all, 26 ng/mL of GAS6 conditioned media was added to the liposomes (or control conditioned media), and the mixture was incubated for 1 h. Serum-starved Hs578T cells were then treated with the two different concentrations of liposomes for 5 and 10 min.

**Stable isotope labeling by amino acids in cell culture**. Hs578T cells were seeded in equivalent amounts for 50% confluency and left it to adhere to plate. Twenty-four hours later, media was changed in one plate with SILAC heavy media (500 mL Arginine and Lysine-free media (Cambridge isotope cat. no. DMEM-500), 10% dialyzed FBS (Invitrogen cat. no. 26400-036), 1% penicillin/streptomycin antibiotics (Wisent), 50 mg Proline (Cambridge Isotopes cat. no. ULM-8333-0.1), 0.4 mM heavy Arginine (Cambridge Isotopes cat. no. CNLM-539-H-0.25), 0.275 mM heavy Lysine (Cambridge Isotopes cat. no. CNLM-291-H-0.25)) and the other with SILAC light media (500 mL Arginine and Lysine-free media (Cambridge isotope cat. no. DMEM-500), 10% dialyzed FBS (Invitrogen cat. no. 26400-036), 1% penicillin/streptomycin antibiotics (Wisent), 50 mg Proline (Cambridge Isotopes cat. no. ULM-8333-0.1), 0.4 mM Arginine (Cambridge Isotopes cat. no. ULM-8347-0.1), 0.275 mM heavy Lysine (Cambridge Isotopes cat. no. ULM-8766-0.1)). Cells were passaged for eight passages for heavy/light amino acid incorporation. The cell labeling was validated via MS to have 99.9% incorporation.

**SILAC sample preparation**. SILAC cells were serum starved in serum-free media for 24 h. Heavy labeled cells were then treated with GAS6 recombinant media for 5, 10, or 20 min, and light labeled cells were treated with control media for the same time points. Cells were lysed with Urea lysis buffer (8 M Urea (Sigma), 20 mM HEPES pH 8.0, 10 mM NaF, 1 mM Na$_4$P$_2$O$_7$, 1 mM NA$_3$VO$_4$, 1× complete protease inhibitor). Lysates were sonicated 3× at 30% amplitude for 30 s and cleared by centrifugation. Proteins were quantified via BioRad DC™ Protein Assay reagents. Equal quantity of proteins was mixed (light and heavy) for each time point. A total of 3 or 20 mg of protein was trypsin digested for the TiO$_2$ enrichment or pY100 immunoaffinity bead immunoprecipitation, respectively.

**Trypsin digestion**. Trypsin digestion was performed by adding 10 μL of trypsin to each sample (1 μg of trypsin [T6567-5 × 20 μg, Sigma] in 200 μL of 20 mM Tris-HCl, pH 8.0). Beads (BioID) and whole lysates (Phosphoproteomics) were incubated at 37 °C on a rotator for ~15–16 h. Next day, 1 μg of trypsin was added again for an additional 2 h of trypsin digestion, and samples were centrifuged for 1 min at 2000 rotations per minute (rpm) at RT. The supernatant was kept in a separate tube, and beads were washed twice with 100 μL of water (8801-7-40, 4 L, Caledon), while each supernatant was pooled with the collected supernatants. Formic acid (94318, 250 mL, Sigma) was added to the solution at a final concentration of 5%. Samples were centrifuged for 10 min at $16,000 \times g$ at RT. The supernatant was transferred and dried in a SpeedVac during 3 h at 30 °C. Samples were then resuspended in 15 μL of formic acid (5%) and stored at −80 °C.

**Phosphopeptide enrichment**. For TiO₂ enrichment, titanium dioxide beads (Canadian Life Science) were resuspended in binding solution (80% Acetonitrile, 3% trifluoroacetic acid (TFA), 290 mg/mL DHB (Sigma) to have a final concentration of 200 μg/μL. Digested peptides are also resuspended in 200 μL of binding solution. Bead slurry (beads + binding solution) is added to the peptide solution in 1:2 ratio of protein:beads (3 mg of peptides:6 mg of beads). Peptide–bead solution is incubated on a rotator for 30 min at RT. Beads were then centrifuged at $5000 \times g$ for 1 min and washed 3× with 75 μL of 30% acetonitrile (ACN) and 3% TFA. Another 3 washes were carried out with 75 μL of 80% ACN and 0.3%TFA. Beads are then eluted 2× with 75 μL of 15% NH₄OH and 40% ACN. Eluted peptides are then dried by speed vacuuming for 2 h. For phosphotyrosine enrichment, PTMScan® Phospho-Tyrosine Rabbit mAB (p-Tyr-1000) Kit (Cell Signaling) was used according to the manufacturer's instructions. TiO₂ enrichment and pY100 immunoaffinity precipitation were done in two biological independent replicates.

**Phosphoproteomics**. Phosphopeptides were analyzed by nano-LC-MS/MS on an Orbitrap Fusion (ThermoFisher Scientific, Bremen, Germany). A 120-min gradient was applied for the LC separation, and standard proteomics parameters were used for a shotgun top 22 analysis and MS3 scanning upon detection of a phosphoric acid neutral loss. Phosphoprotein identifications and phospho-modulation analyses were performed using MaxQuant[50] (version 1.6.1.0) against the human RefSeq protein database (version July 29, 2016), and we considered as modulated phosphosites showing a normalized Log2 (Avg (H/L ratio)) ≤ −0.5 or ≥0.5 at either 5, 10, or 20 min. We compared our AXL phosphoproteomic data to a combination of two EGFR phosphoproteomic studies[20,21]. This comparison was performed on AXL and/or EGFR phospho-modulated proteins showing a normalized Log2 (Avg (H/L ratio)) ≤ −0.5 or ≥0.5 at either time points.

**Proximity-dependent BioID**. BioID experiments were carried out as described previously[51]. Briefly, Flp-In™ T-REx™ 293 cells were engineered to express BirA*-Flag-NEDD9, BirA*-Flag-PEAK1 WT, BirA*-Flag-PEAK1 3PA, or the control BirA*-Flag-EGFP in a tetracycline-inducible manner (1 μg/mL) by transfecting Flp-In™ T-REx™ 293 cells with 2 μg of pOG44 and 500 ng of BirA*-Flag-NEDD9 or BirA*-Flag-PEAK1 using Lipofectamine 2000 (Invitrogen). Cells were then selected by Hygromycin B for 12 days. Positive clones were grown in two 15-cm plates prior to treatments with tetracycline and biotin (50 μM). After 24 h of treatments, cells were harvested at ~90–95% confluence. The medium was discarded, and the cells were washed with 5 mL ice-cold PBS and scraped from the plates to be transferred into 15-mL tubes. Tubes were centrifuged during 5 min at $500 \times g$ at 4 °C. This washing step was repeated two times, and cell pellets were kept at −80 °C. Later, a 50-mL stock of RIPA buffer was prepared and supplemented with 500 μL of 100 mM phenylmethanesulfonylfluoride (P7626-1G, Sigma), 50 μL of 1 M dithiothreitol (15500813, 5 g, Thermo Fisher), and 100 μL of protease inhibitor (P8340, 1 mL, Sigma). Cell pellets were thawed, resuspended in 1.5 mL of supplemented RIPA buffer and 1 μL of benzonase (71205-3, 250 U/μL, EMD Millipore) was added into each sample. Samples were sonicated three times during 30 s at 30% amplitude with 10 s bursts and 2 s of rest in between. Samples were then centrifuged for 30 min at $16,000 \times g$ at 4 °C. After centrifugation, 20 μL of supernatant were kept aside to evaluate the levels of protein expression and biotinylation by immunoblotting. The remaining lysate was incubated with 70 μL of pre-washed streptavidin beads (17-5113-01, 5 mL, GE Healthcare) during 3 h on a rotator at 4 °C. Samples were centrifuged for 1 min at 2000 rpm at 4 °C, and the supernatant was discarded. Beads were washed three times with 1.5 mL of RIPA buffer and centrifuged during 1 min at 2000 rpm at 4 °C. Beads were washed three times by resuspending in 1 mL of 50 mM ammonium bicarbonate (AB0032, 500 G, Biobasic) and centrifuged for 1 min at 2000 rpm and 4 °C. Samples were trypsin-digested (see "Trypsin digestion" method) and injected into an Orbitrap Velos Mass Spectrometer (Thermo Fisher). Peptide search and identification were processed using the Human RefSeq database (version 57) and the iProphet pipeline[52] integrated in Prohits[53]. The search engines were Mascot and Comet, with trypsin specificity and two missed cleavage sites allowed. The resulting Comet and Mascot search results were individually processed by PeptideProphet[54], and peptides were assembled into proteins using parsimony rules first described in ProteinProphet[55] into a final iProphet[52] protein output using the Trans-Proteomic Pipeline. For each duplicated bait, we used SAINTexpress (version 3.6.1,

nControl:4, fthres:0, fgroup:0, var:0, nburn:2000, niter:5000, lowMode:0, minFold:1, normalize:1, nCompressBaits:2) on proteins with iProphet protein probability ≥0.85 with unique peptides ≥2 and considered statistically significant NEDD9 or PEAK1 interactors displaying a BFDR threshold ≤0.02 against 4 biological replicates of the BirA*-Flag-EGFP control. Prey's abundance were normalized by applying the SAF[56] (Spectral Abundance Factor) method. The SAF metric was calculated by dividing the protein's spectral count on the protein's length (from Uniprot). We used ProHits-viz[57] to generate dot plot analyses.

**Bioinformatics analyses**. All proteomics data were imported into a local MySQL database. Graphical representations of protein–protein networks were generated with Cytoscape[58] (version 3.6.1) using the application GeneMANIA[59] (version 3.4.1 and its human database version 13 July 2017) or the built-in PSICQUIC client by merging networks generated from Intact, Mint, Reactome, and Uniprot databases. Each protein was annotated with the Gene Ontology by importing the Gene Ontology Annotation Database[60] and our phospho-modulation values in order to identify relevant phospho-modulated functions. We also identified phosphatase and kinase families by importing the Phosphatome[61] and Kinome[62] classifications. Functional enrichment analyses of the identified interactors were assessed with g:Profiler[63] against the Gene Ontology and KEGG[64] databases with moderate hierarchical filtering and by applying statistical correction with the Benjamini–Hochberg false discovery rate method. Dot plots of overrepresented KEGG pathways (pvalueCutoff = 0.05) were generated with the ClusterProfiler[65] package in R (www.r-project.org).

**Production of PEAK1 KO cells via CRISPR/CAS9-mediated gene targeting**. MDA-MB-231 cells were infected with the CAS9 plasmid cited in Supplementary Table 3 and selected with 10 μg/mL Blasticidin for 10 days. MDA-MB-231-CAS9 cells were then infected separately with three different sgRNA plasmids described[66] in Supplementary Table 3. These cells were selected with 1 μg/mL Puromycin for 3 days and were isolated in single cells in 2 96-well plates. Single clones for each knockout were grown to maintain a homogenous pool of PEAK1 knockout cells. After testing multiple clones, we chose two clones for each knockout to further validate them. Cells that survived and grew colonies were tested for their efficiency of PEAK1 deletion via western blotting. To generate cells that are Luciferase positive, we infected the MDA-MB-231-CAS9 control and MDA-MB-231-CAS9 PEAK1 knockout cells with a luciferase plasmid mentioned in Supplementary Table 3. Cells were selected with 500 μg/mL Hygromycin for 5 days. For animal experiments, two clones (A and B) of two (KO1 and KO2) out of the three different PEAK1 sgRNA knockout cells were mixed to create PEAK1 knockout pools (KO1 and KO2) to increase the heterogeneity of the model and decrease the specific off-target effects by the different sgRNA.

**Generation of CRISPR-resistant cDNAs to rescue PEAK1 KO cells**. To generate a CRISPR-resistant cDNA, pRetroX-Sgk269 WT was used to mutate the PAM site at the end of the sgRNA from NGG to NGA using site-directed mutagenesis. This was done for pRetroX-Sgk269 WT and pRetroX-Sgk269 3PA. Once generated, MDA-MB-231-CAS9 PEAK1 knockout cells (KO1) were retrovirally infected with pRetroX (EV), pRetroX-Sgk269 WT PAM mutant (PEAK1 WT), or pRetroX-Sgk269 3PA PAM mutant (PEAK1 3PA). After few days of infection, flow cytometry was used to sort for DsRED-positive cells, which was a marker found in the backbone of pRetroX plasmid. DsRED-positive cells were validated via western blotting.

**Immunofluorescence confocal microscopy**. Cells plated on glass coverslips were fixed with 4% paraformaldehyde (PFA) and permeabilized in 0.1% Triton in PBS. Cells were blocked with 2% BSA in 0.1% Triton and incubated with the indicated antibodies for 1 h at RT. Cells were then washed 3 times with 0.1% Triton in PBS and incubated with the corresponding secondary antibody for 30 min at RT. Following this incubation, cells were washed again 3 times with 0.1% Triton in PBS and stained for Phalloidin using the indicated antibody in Supplementary Table 1 for 30 min at RT. Cells were washed 3 additional times with 0.1% Triton in PBS and coverslips were mounted on slides using SlowFade Gold reagent (Invitrogen). Pictures were acquired with Zeiss LSM710 confocal microscope at objective ×63. Experiments were done in triplicates, and 15 cells were used per condition per experiment for quantifications.

**Proximity ligation assay**. Cells plated on glass coverslips were fixed, permeabilized, blocked, and incubated with the indicated antibodies as mentioned in immunofluorescence. PLA was then performed using the DuoLink In Situ PLA Detection Kit (Sigma). Hybridization with PLA probes (plus and minus) for rabbit and mouse, ligation, and amplification of the PLA signal were performed according to the manufacturer's instructions. Images were taken using Zeiss LSM710 confocal microscope at objective ×63. PLA signal was quantified per cell using the ImageJ software. PLA signal is depicted in green and Phalloidin in red. Experiments were done in triplicates, and 15 cells were used per condition per experiment for quantifications.

**Boyden migration and invasion assay**. Boyden assays were performed using 8 μm pores Boyden Chambers (24-well, Costar). For the invasion assays, upper chamber was coated with 6 μL of Matrigel (BD Biosciences) dissolved in 100 μL of DMEM. Cells were detached and washed with DMEM 0.1% BSA. One hundred thousand cells were seeded in the top chamber and allowed to migrate for 6 h (migration) or 16 h (invasion) toward the bottom chamber containing 10% FBS. Upper and lower chambers were then washed with 1×PBS and cells on bottom side of the chamber were fixed with 4% PFA. Cells in the upper chambers were removed using cotton swabs, and the membrane was mounted on a glass slide using SlowFade Gold reagent (Invitrogen). The average number of migrating cells in 10 independents ×20 microscopic fields were evaluated, and each experiment was performed in triplicates.

**MTT (3-[4,5-dimethylthiazol-2-yl]-2,5 diphenyl tetrazolium bromide) proliferation assay**. MDA-MB-231 cells were plated in 96-well plate 5000 cells/100 μL of regular culture media in 5 replicates (wells) for each cell type. Five 96-well plates were plated for 5 different days of reading. MTT reaction mix (2.5 mM MTT (Invitrogen), 15 nM HEPES, red-phenol-free and FBS-free DMEM) was added to each well and incubated for 4 h at 37 °C for MTT incorporation. To stop the MTT reaction, stop solution (2% dimethyl sulfoxide, 0.1 M Glycine pH11) was added to each well for 5 min, and the plate was read in an ELISA reader at 540 nm wavelength. For the different day reading, MTT reaction mix was added to the plate at the same time as the previous days for consistency. This experiment was performed in three independent replicates.

**Annexin V-FITC apoptosis staining**. Hs578T cells grown in either 10% serum or serum-starved conditions and serum-starved GAS6-His Hela cells treated with −/+tetracyline were tested for apoptosis using the Annexin V-FITC Apoptosis Staining/Detection Kit (Abcam). Briefly, cells in suspension were stained for Annexin V and PI for 5 min in the dark, and stained cells were quantified by flow cytometry. Annexin V-FITC binding and PI staining were analyzed using signal detector FL1 and FL2, respectively.

**Tumorsphere-formation assay**. MDA-MB-231 cells were plated in low adherence in DMEM/F12 media supplemented with 0.4% FBS, EGF (20 ng/mL), fibroblast growth factor (10 ng/mL), insulin (5 μg/mL), and B27 supplement (Invitrogen 17504-044) as described in Lo et al.[67]. One week later, the quantification of tumorspheres formed was conducted using a DM IRE2 microscope (Leica).

**Immunoprecipitation, GST-pulldown assay, and immunoblotting**. Cells were lysed with either RIPA (50 mM Tris pH7.6, 0.1% sodium dodecyl sulfate (SDS), 0.5% sodium deoxycholate, 1% Nonidet P-40, 5 mM EDTA, 150 mM NaCl, 10 mM NaF, 1 mM $Na_4P_2O_7$, 1 mM $NA_3VO_4$, 1× complete protease inhibitor) or 1% NP-40 (15 mM NaCl, 50 mM Tris pH7.5, 1% Nonidet P-40, 10 mM NaF, 1 mM $Na_4P_2O_7$, 1 mM $NA_3VO_4$, 1× complete protease inhibitor) buffer, and cell lysates were cleared by centrifugation. Protein quantification was performed using $DC^{TM}$ Protein Assay reagents (BioRad). For immunoprecipitations, 1 mg of protein lysate was incubated with the indicated antibody and corresponding agarose beads (Protein A or G) for 3 h at 4 °C. Immunoprecipitates were then washed with the corresponding lysis buffer and denatured in 6× lysis buffer. For GST pulldowns, 1 mg of protein lysate was incubated with the corresponding GST beads for 2 h at 4 °C. Lysates, immunoprecipitates, and GST complexes were run on SDS–electrophoresis acrylamide gels at 180 V and transferred on nitrocellulose or polyvinylidene difluoride for 3 h at 4 °C at 50 V or overnight at 4 °C at 20 V. Immunoblots are then blocked with 1% BSA and incubated with the indicated primary antibodies, mentioned in Supplementary Table 1, overnight at 4 °C or RT. Immunoblots are then washed with 0.01% TBST three times and incubated with the corresponding secondary antibody for 30 min at RT. Protein signals are then revealed via Clarity$^{TM}$ western ECL substrate (BioRad). Uncropped and unprocessed scans of all blots are supplied in Supplementary Fig. 8.

**GST protein purification**. A strike of the glycerol stock of the GST protein is grown in LB with antibiotics overnight at 37 °C. Twenty-four hours later, the bacteria culture is grown in 5× the volume of LB with antibiotics and 0.1 M IPTG. This culture is then put to shake at 37 °C for 3 h. Once the pellet is obtained by centrifugation, 4 mL of lysis buffer (1× PBS, 1% Triton, 1× complete protein protease inhibitor) was used to lyse the cells. After rupturing the cell membranes by sonication 3× of 30 s, a cell debris pellet is obtained by centrifugation. The supernatant containing the GST protein is incubated with GST beads for 1 h to purify the GST-tagged protein. The GST beads are then washed and resuspended in 0.1% PBS Triton. To purify the GST protein, GST protein is run on a column and is eluted with 10 aliquots of elution buffer (10 mM Glutathione, 50 mM Tris pH 8.0). Eluates are then quantified by Quick Start$^{TM}$ Bradford Dye Reagent (BioRad). Concentrated elutes containing protein are pooled and placed into a centrifugal filter unit and centrifuged for 10 min at 4000 rpm. Purified protein is collected and stocked at −80 °C.

**Live cell imaging**. MDA-MB-231 cells were transfected with 1.5 μg of GFP-PXN. Twenty-four hours later, cells were plated on 35-mm fibronectin-coated glass bottom plates (MatTek Corporation). Once adhered, cells were either serum starved for GAS6 stimulation or treated with AXL inhibitor R428 for 1 h at 1 μM. Cells were then imaged while incubated with GAS6 or R428 using Carl Zeiss Spinning Disk Confocal microscopy and ZEN imaging program at 1-min intervals for 30 min, using the EGFP laser (488 nm) at 3% strength. Videos and images were obtained using IMARIS 8.0 and analyzed via the Image J software. Lifetime of the FAs was measured from the time an FA appeared to the time it disappeared. FA assembly rate is the time an FA takes to increase in size and intensity. Disassembly time is the time it takes an FA intensity to decrease. Cells were performed in triplicates with 90 FAs followed per condition per experiment[68].

**Animal experiments**. All animal experiments were approved by the Animal Care Committee of the Institut de Recherches Cliniques de Montréal and complied with the Canadian Council of Animal Care guidelines. Tail veins and graft experiments were conducted in athymic nude NU/J mice obtained from Jackson Laboratories in a specific pathogen-free facility. Mice were kept with a light cycle of 12-h light/12-h dark cycle. Temperatures of 65–75 °F (~18–23 °C) with 40–60% humidity were also used as housing conditions for the mice.

**Experimental metastasis, mammary fat pad grafts, and bioluminescence imaging**. Tail veins and fat pad grafts were conducted as previously described in ref. [11]. Briefly, for experimental metastasis assay, $10^6$ cells resuspended in PBS were injected in the lateral tail vein of 6–8-week-old Nude mice. For fat pad grafts, $10^6$ cells were injected in the mammary fat pad of 3-week-old Nude mice. Xenogen IVIS 200 (PerkinElmer) and Living Image 4.2 software was used for bioluminescence imaging. In all, 150 mg/kg of Beetle Luciferin (Promega) solution (stock of 15 mg/mL in PBS) was injected intraperitoneally 10 min before imaging, and photon flux was calculated for each mouse using a region of interest.

**Blood burden assay (CTCs)**. The quantification of CTCs was performed as previously described[11]. Briefly, blood was drawn via heart puncture (0.5 mL) with a 25-gauge needle and a 1-mL syringe coated with heparin (100 U/mL). Blood was plated in DMEM/F12 20% FBS and media was changed every 2 days. After 8 days, tumor cell colonies in the dish were counted, and the tumor blood burden was calculated as the total colonies divided by the volume of blood taken as described in ref. [69].

**Statistics and reproducibility**. Phosphoproteomic data were done in duplicates (twice for $TiO_2$ enrichments and twice for pY100 immunoaffinity enrichment). Data are presented as mean ± SEM from at least three independent experiments. Statistical analyses were performed with the GraphPad Prism Software using the two-tailed Mann–Whitney non-parametric $t$ test with unpaired $t$ test with a two-tailed $P$ value with 95% confidence interval. All in vitro experiments were performed in three independent experiments with similar results, and the representative experiments are shown in the figure panels.

**Reporting summary**. Further information on research design is available in the Nature Research Reporting Summary linked to this article.

## Data availability

The raw proteomics data, which are presented in Figs. 1, 2, 4, and 5, are uploaded to the MassIVE archive [http://proteomecentral.proteomexchange.org/cgi/GetDataset?ID=PXD018582]. The raw western blots underlying Figs. 1b, e, 4a–c, f, 5b, c, 6b–d, g, 7a–e, 8c, 9a, and 10a and Supplementary Figs. 1a, b, f, 4b–d, g, h, 5a–d, g, 6a, f, and 7l, m are provided in Supplementary Fig. 8 found in Supplementary Information file. All databases used in this study are mentioned in the "Methods" section under "Bioinformatics analysis." All cell lines and constructs generated in the manuscript are available upon requests from authors. The remaining data are available within the article, Supplementary Information, or available from the authors upon reasonable request. Source data are provided with this paper.

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

## Acknowledgements
We thank Dr. Elena Pasquale and Dr. Kristiina Vuori for critical reading of the manuscript. We thank Dr. Liliana Attisano, Dr. Christopher E. Turner, Dr. Ed Manser, and Dr. Erica Golemis for providing plasmids. We thank Dr. Nadia Dubé, Dr. Amélie Robert, and Dr. Philippe Roux for insightful discussions. We thank Dr. Denis Faubert in the IRCM Proteomics facility for guidance and for the processing of MS samples and Dr. Dominic Filion for microscopy assistance. We thank Stephen Taylor (U of Manchester) for Flp-In™ T-REx™ HeLa cells. This work was supported by operating grants from the Canadian Institute of Health Research (MOP-142425 to J.-F.C. and J.-P.G.) and National Science and Engineering Research Council of Canada (RGPIN-2016-04808 to J.-F.C.). A.A. and H.B. are recipients of FRQS Doctoral studentship and M.-A.G. is supported by a CIHR studentship. J.-F.C. holds the Transat Chair in Breast Cancer Research and was supported by FRQS Senior investigator career award. R.J.D. was supported by a fellowship (APP1058540) and Project Grant (APP1129794) from the National Health and Medical Research Council of Australia.

## Author contributions
A.A.-T., M.-A.G., M.F., J.-P.G., and J.-F.C. designed the research; A.A.-T., M.-A.G., C.D., C.A., C.S., H.B., and M.-P.T. performed the research; D.D., A.V., and R.J.D. provided unique reagents; A.A.-T., M.-A.G., J.B., C.D., C.A., C.S., R.C., H.B., M.F., A.-C.G., and J.-F.C. analyzed the data; A.A.-T. and J.-F.C. wrote the paper with input from all other authors.

## Competing interests
The authors declare no competing interests.
