## [Peer Review File · Nature Communications]

Reviewers' comments:

Reviewer #1 (Remarks to the Author):

This is an involved study that implicates receptor tyrosine kinase organized by the enzyme AXL as a potential mechanism to enhance metastasis and acquisition of resistance to cancer drugs. This is a very complicated study with many parts that is compounded by the frequent reference to supplemental data. Some of which should be omitted and some of which should be included in the body of the text. Using a variety of indirect methods, the authors propose that AXL signaling contributes to FA dynamics. This work could benefit from more mechanistic experiments that clearly define the role of AXL and the pseudokinase PEA1 in tumor growth and metastasis. More specific criticisms are listed below.

On page 4 it states "We identified an overrepresentation of MAPK, CDK, PKA and PKC kinase motifs in most of the clusters, suggesting that AXL may signal via regulation of these kinases (Supplementary Information, Fig. S2d). This data is weak and extremely speculative. Since it covers several of the predominant kinases it is hard to know what this means. I recommend removal of this data.

The proximity ligation data presented in fig 2a is poor and hard to evaluate. Likewise, The data in figures 2b and c is not very convincing. First, quantitation of the PLA data should and Quantitation of the number of focal adhesions as dot scatter plots is a more compelling way to show the relative effect of changing AXL expression or action.

Evidence for changes in the number of focal adhesions presented in figure 2c and the supplementary information is far from convincing.

The Bio ID proteomic data in figures 3f and g is poorly connected to the rest of the paper. This is a rather speculative element of the work that needs some more experimental evidence to warrant its inclusion in the study.

Reviewer #2 (Remarks to the Author):

This manuscript offers a fascinating and important study of AXL signaling, which is timely because of the growing appreciation of the key role AXL plays in tumor resistance to targeted and chemotherapeutic drugs. The kind of understanding being pursued here, of nodes downstream of AXL activation could be very useful in suggesting effective combination treatments as well as in gaining insights into novel receptor tyrosine kinase signaling biochemistry. The authors bring together multiple approaches, experimental and computational, to ascertain relevant pathway elements including NEDD9, PEA1, CSK, and Paxillin, toward enhancing invasive tumor cell migration via focal adhesion turnover. The resulting story is appealing and appears to be generally sound.

My main point of concern relates to the technical details of Gas6 treatment for AXL stimulation. The authors seem to be not familiar with certain complications inherent in the Gas6/AXL system, such that mere treatment with soluble Gas6 is well-known to have variable effects depending on availability and characteristics of lipid vesicle-mediated interactions (for instance, as studied by Meyer et al in Cell Systems [2015]), such that AXL activation effects will vary with factors that are not explicitly considered in the experimental protocols outlined here. I believe that the authors need to provide more detailed characterization of their Gas6 protocols, including whether the cells being treated express Gas6 themselves in autocrine manner and to what degree apoptotic cell debris might be present due to low levels of nonviability in the culture conditions. These details will

be vital for other laboratories in any attempts to reproduce and build upon the findings here.

Reviewer #3 (Remarks to the Author):

In this paper the authors try to elucidate the major signaling pathways activated by the receptor tyrosine kinase, AXL.

The authors have used systems biology and proteomic approaches and find that AXL modulates the phosphorylation of a network of focal adhesion (FA) proteins that culminates in faster FA disassembly. They go on to delineate a rather convoluted pathway whereby AXL phosphorylates the FA protein NEDD9, which then promotes NEDD9/CRKII-mediated phosphorylation of PEA1. This pseudokinase then complexes with CSK kinase to phosphorylate paxillin, resulting in FA turnover.

The strengths of this work lie in the biochemistry and systems biology, which are carried out very well. The finding that a major network of AXL-mediated phosphorylation events occur in FA proteins is novel, and this work does suggest that AXL can modulate FA turnover.

The weaknesses lie in the biology and physiological relevance of the findings to tumor progression, especially the relevance of focal adhesion turnover-per se to AXL-mediated metastasis.

Thus while the final data in Figs 5 and S7 using PEA1 KO MDA-MB-231 cells suggest that PEA1 may contribute to metastasis, The authors have not connected the pathway back to AXL. Thus, PEA1 may have completely AXL-independent role in regulating metastasis. To make the case that it is the FA turnover pathway

mediated by AXL that promotes metastasis, the authors need to demonstrate that some of the key phosphorylation and protein-protein interactions leading PEA1-mediated to paxillin phosphorylation and FA turnover also promote metastasis.

Otherwise the paper does not hang together very well as it represents some excellent biochemistry and a separate finding that PEA1 KO modulates metastasis.

Specific Points that should be addressed:

1. Introduction and Discussion sections are missing. These need to be added to provide a context to this work and to demonstrate originality of the work
2. All the phospho-blot images lack blots of the total proteins. This is essential to show that it is the phosphorylation of the protein and not the total expression of the proteins that is changing.
3. In Fig 2 the analyses of the FAs should also include effects of GAS6 induced effects in addition to the constitutive situation, i.e. effects of GAS6 + R428.
4. p12. Does inhibition of paxillin phosphorylation (directly or by using phospho-mutants) have the same effect on metastasis as PEA1 KO?
5. The effect of AXL KO or the effect of AXL inhibitor on metastasis has to be carried out. Is this effect rescued by PEA1?
6. The metastasis data (PEA1 KO) need to be carried out in another TNBC cell line in addition to MDA-MB-231 cells
- 7 Fig S7j, why were the lungs probed ex vivo. Could in vivo signal be detected?
8. Fig S7d/e: Tumor volumes should be measured , not only IVIS signal.
9. The pathway model should be depicted in the main figures.

We would like to thank the reviewers for their expert comments on our work. You'll find below a point-by-point response.

REVIEWER 1.

“This is an involved study that implicates receptor tyrosine kinase organized by the enzyme AXL as a potential mechanism to enhance metastasis and acquisition of resistance to cancer drugs. This is a very complicated study with many parts that is compounded by the frequent reference to supplemental data. Some of which should be omitted and some of which should be included in the body of the text. Using a variety of indirect methods, the authors propose that AXL signaling contributes to FA dynamics. This work could benefit from more mechanistic experiments that clearly define the role of AXL and the pseudokinase PEAK1 in tumor growth and metastasis. More specific criticisms are listed below.”

We thank the reviewer for his comments. Our manuscript was initially assessed at *Nature Cell Biology* where we opted to submit it in a “Letter” format. The paper was next transferred to *Nature Communications* in the same format and sent in review by the editor. We believe that most of the irritants and organizational issues were fixed when we reformatted the paper in the *Nature Communications* style. We are re-submitting a profoundly re-organized manuscript. We are confident that the paper is significantly clearer in the revised longer format of *Nature Communications*.

Regarding the mechanistic experiments, please see our answers to Reviewer 3 who raised a very similar comment.

“On page 4 it states “We identified an overrepresentation of MAPK, CDK, PKA and PKC kinase motifs in most of the clusters, suggesting that AXL may signal via regulation of these kinases (Supplementary Information, Fig. S2d). This data is weak and extremely speculative. Since it covers several of the predominant kinases it is hard to know what this means. I recommend removal of this data.”

We appreciate this comment. We believe that these data may have contributed to overload the manuscript. While we trust that there are interesting findings in that figure, we recognize that further validation is needed. As such, we will keep these data for a future study as this is not essential for the main message of this manuscript.

“The proximity ligation data presented in fig 2a is poor and hard to evaluated. Likewise, the data in figures 2b and c is not very convincing. First, quantitation of the PLA data should and Quantitation of the number of focal adhesions as dot scatter plots is a more compelling way to show the relative effect of changing AXL expression or action.” “Evidence for changes in the number of focal adhesions presented in figure 2c and the supplementary information is far from convincing.”

We now provide better images of the PLA experiments and modified their quantitation as requested. In particular, for Figure 2c and d, experiments were repeated to obtain more representative images that also now include actin (phalloidin) staining. We also provide the quantitation as scatter plots. The data shown in Figure 2 c-d was highly reproducible and is also supported by videos of cells expressing GFP-Paxillin in the paper that are even more visual. Globally, we are confident that these data are now convincing.

“The Bio ID proteomic data in figures 3f and g is poorly connected to the rest of the paper. This is a rather speculative element of the work that needs some more experimental evidence to warrant its inclusion in the study.”

We were rather surprised by this comment. NEDD9 was one of the most significantly phosphotyrosine-modulated proteins in our screen and therefore a logical candidate to follow up. As far as experimental

validations, it is important to note that we set up stringent bioinformatics analysis parameters for the BioID data. Of course, we could attempt to validate several candidates, but it is unlikely that we would follow up on more of them given that our story is already quite broad. Instead, this experiment gave us insight into the proximity proteins to NEDD9 and was key for us to identify PEAK1 as a recurrent protein from all 3 screens (pY and pS/T proteomic and BioID Nedd9) for follow up mechanistic studies. In conclusion, and in light of the insightful reviewer comment, we now believe that we have introduced a significantly improved rationale for these experiments in the revised manuscript.

REVIEWER 2

“This manuscript offers a fascinating and important study of AXL signaling, which is timely because of the growing appreciation of the key role AXL plays in tumor resistance to targeted and chemotherapeutic drugs. The kind of understanding being pursued here, of nodes downstream of AXL activation could be very useful in suggesting effective combination treatments as well as in gaining insights into novel receptor tyrosine kinase signaling biochemistry. The authors bring together multiple approaches, experimental and computational, to ascertain relevant pathway elements including NEDD9, PEAK1, CSK, and Paxillin, toward enhancing invasive tumor cell migration via focal adhesion turnover. The resulting story is appealing and appears to be generally sound.”

We thank this reviewer for recognizing the importance of our timely study. We carried out this work exactly in that spirit.

“My main point of concern relates to the technical details of Gas6 treatment for AXL stimulation. The authors seem to be not familiar with certain complications inherent in the Gas6/AXL system, such that mere treatment with soluble Gas6 is well-known to have variable effects depending on availability and characteristics of lipid vesicle-mediated interactions (for instance, as studied by Meyer et al in Cell Systems [2015]), such that AXL activation effects will vary with factors that are not explicitly considered in the experimental protocols outlined here. I believe that the authors need to provide more detailed characterization of their Gas6 protocols, including whether the cells being treated express Gas6 themselves in autocrine manner and to what degree apoptotic cell debris might be present due to low levels of nonviability in the culture conditions. These details will be vital for other laboratories in any attempts to reproduce and build upon the findings here.”

This is a perfectly valid point. However, it does not stem from our unfamiliarity of the AXL/GAS6 field. Similar to our answer to Reviewer 1, the manuscript was written in a Letter format. In fact, this information was present in our initial drafts but was deleted to meet the character requirements of *Nature Cell Biology*. We have now discussed this clearly in the revised manuscript.

We completely agree that the form in which GAS6 is presented to AXL expressing cells has the potential to change the biological responses. We have focused on the model of soluble GAS6 presented to AXL as a manipulated system to conduct the first phospho-screens for this kinase. During the revision process, we coated liposomes with 2 different concentrations of GAS6 and tested their efficiency in activating AKT in comparison to GAS6 conditioned media that we used throughout the study. As discussed in the paper, GAS6 presented on liposomes activated AKT to the same extent as GAS6 conditioned media (Fig S1f). We also confirmed that there is no increase in apoptotic cell content in our conditioned media or as a result of our starvation protocol of Hs578T cells (Fig S1d, e). We believe that these additional experiments addressed this important point raised by the reviewer. We nevertheless agree that our work will open the door to new screens that should be conducted with different means of AXL stimulation, including exposure to apoptotic cells coated by GAS6.

We can re-assure this reviewer that control cells are critical for our proteomics screens. Even if some apoptotic debris were to be present, we are focusing on the fold changes between unstimulated and GAS6-stimulated conditions. In support of the argument that apoptotic cells are not responsible for the activation of AXL following serum starvation, we do not find any pY-AXL in control conditions while pY-AXL and pAKT are robustly increased following GAS6 addition (Fig 1b).

We would also like to mention that Meyer et al. largely used GAS6 from R&D. We find this GAS6 to be poorly γ -carboxylated in contrast to the one we prepared here. As shown in Figure S1b, our soluble GAS6 is highly γ -carboxylated, which we found to be sensitive to warfarin treatment in agreement with the literature. Accordingly, the R&D GAS6 poorly activates AXL when added to the media. In the Meyer study, inclusion of GAS6 in liposomes may rescue the poor activity of this ligand and reveal a level of activity that we determined with our GAS6 conditioned media. Hence, one has to be careful regarding the interpretation of the data obtained with GAS6 from R&D.

REVIEWER 3

“In this paper the authors try to elucidate the major signaling pathways activated by the receptor tyrosine kinase, AXL. The authors have used systems biology and proteomic approaches and find that AXL modulates the phosphorylation of a network of focal adhesion (FA) proteins that culminates in faster FA disassembly. They go on to delineate a rather convoluted pathway whereby AXL phosphorylates the FA protein NEDD9, which then promotes NEDD9/CRKII-mediated phosphorylation of PEA1. This pseudokinase then complexes with CSK kinase to phosphorylate paxillin, resulting in FA turnover.”

“The strengths of this work lie in the biochemistry and systems biology, which are carried out very well. The finding that a major network of AXL-mediated phosphorylation events occur in FA proteins is novel, and this work does suggest that AXL can modulate FA turnover.”

We thank the reviewer for the positive comments on our paper. Regarding the convoluted aspect of the pathway, we strongly believe that we now present a clearer story.

“The weaknesses lie in the biology and physiological relevance of the findings to tumor progression, especially the relevance of focal adhesion turnover-per se to AXL-mediated metastasis. Thus while the final data in Figs 5 and S7 using PEA1 KO MDA-MB-231 cells suggest that PEA1 may contribute to metastasis, the authors have not connected the pathway back to AXL. Thus, PEA1 may have completely AXL-independent role in regulating metastasis. To make the case that it is the FA turnover pathway mediated by AXL that promotes metastasis, the authors need to demonstrate that some of the key phosphorylation and protein-protein interactions leading PEA1-mediated to paxillin phosphorylation and FA turnover also promote metastasis. Otherwise the paper does not hang together very well as it represents some excellent biochemistry and a separate finding that PEA1 KO modulates metastasis.”

Note: this comment is identical to a comment of Reviewer 1.

This is an excellent comment. We addressed this comment by conducting additional experiments in vitro and in vivo on PEA1 expression to strengthen the connection to AXL signalling. For this, we compared the expression of PEA1 WT and a PEA1 mutant (proline-rich domain; PEA1 3PA – see Fig 6a-c and Fig 7e) that we demonstrated in this study to be completely uncoupled from AXL-induced phosphorylation, cannot recruit CRKII/NEDD9 complex and is unable to induce paxillin phosphorylation. Hence, we rescued PEA1 knockout cells with CRISPR/Cas9 resistant versions of either PEA1 wildtype or PEA1 3PA mutant (Fig 10a).

By performing cell-based assays, we found that the PEAK1 3PA mutant led to less migration, invasion and wound healing (Fig S7g-i) in comparison to PEAK1 wildtype. In addition, the expression of the PEAK1 3PA mutant in PEAK1 KO cells led to the formation of fewer, but larger adhesions when compared with the expression of PEAK1 WT (Fig 10f-h). These data demonstrate that the link between AXL/PEAK1 complex is needed for promoting FA turnover and migration/invasion.

These experiments were further validated *in vivo* in a metastasis assay. We found that PEAK1 WT and the 3PA mutant both rescued the tumor growth defect observed for the PEAK1 KO cells when grafted in mammary fat pads of Nude mice (Fig 10b-c, Fig S7j). While re-expression of PEAK1 WT rescued spontaneous metastasis and increased the number of circulating tumor cells (Fig 10d-e), the PEAK1 3PA mutant cells behaved similarly to the PEAK1 KO cells and failed to form metastases. Hence, these data support a model where the AXL/PEAK1 connection is required for spontaneous metastasis but not tumor growth. This study not only provides mechanistic insights into AXL signalling, but also provides the first demonstration that PEAK1 is required for TNBC metastasis and it demonstrates that PEAK1/CRK coupling and AXL-mediated phosphorylation of PEAK1 is critical for this process.

We believe these data further address the reviewer's concerns by connecting AXL to PEAK1 and therefore tie our findings together.

“Specific Points that should be addressed:

“1. Introduction and Discussion sections are missing. These need to be added to provide a context to this work and to demonstrate originality of the work.”

As stated above, this manuscript was initially submitted in a “Letter” format. This is now corrected.

“2. All the phospho-blot lack blots of the total proteins. This is essential to show that it is the phosphorylation of the protein and not the total expression of the proteins that is changing.”

While it is very unlikely that the levels of the signalling intermediates are changing in 5-20 minutes timepoints assayed in these experiments, we nevertheless obtained primary antibodies of these targets to further normalize the data. All phospho-blot are accompanied with their total blots (Fig 1e).

“3. In Fig 2 the analyses of the FAs should also include effects of GAS6 induced effects in addition to the constitutive situation, i.e. effects of GAS6 + R428.”

This experiment was addressed and included in Fig 3c. Addition of R428 to GAS6 treated cells reversed the effect of GAS6 on FA number.

“4. p12. Does inhibition of paxillin phosphorylation (directly or by using phospho-mutants) have the same effect on metastasis as PEAK KO?”

This is an interesting comment pointing to some of the limitations of the work we present here. For example, the screens are not capable of identifying all phosphorylation events. Also, they are not capable of identifying phosphosites that fall in regions that are not efficiently digested by trypsin. Hence, they represent a snapshot of the ongoing signaling. Currently, we have not mapped all of the sites on paxillin where phosphorylation is increased by PEAK1. We can certainly say Y31 and Y118, but there may be others. This complicates the interpretation of the data from the requested experiment. We believe that this point is important and beyond the scope of the current study. A study dedicated to mapping AXL-PEAK-mediated phosphorylation of Paxillin and the functional significance would merit its own study.

“5. The effect of AXL KO or the effect of AXL inhibitor on metastasis has to be carried out. Is this effect rescued by PEAK?”

This comment is difficult to interpret. We (and others) have done *in vivo* treatments of R428¹ and it indeed blocks metastasis. It is not clear that PEAK1 over-expression would rescue AXL inhibition as our model demonstrates that phosphorylation events are required (including AXL-induced phosphorylation of PEAK1). We also fear that overexpression of PEAK1 may have dominant pro-migration/invasion effects unrelated to AXL, and as such, it would be difficult to interpret this experiment, especially to link AXL-PEAK-PXN.

“6. The metastasis data (PEAK KO) need to be carried out in another TNBC cell line in addition to MDA-MB-231 cells.”

Our attempt to inactivate PEAK1 in Hs578T TNBC model cell line by CRISPR was not successful for technical reason (i.e. cell survival upon clonal dilution). However, to generalize the importance of PEAK1, depletion of PEAK1 by siRNA was shown to decrease metastasis and tumor growth in a K-Ras pancreatic cancer model². We also provide evidence that we can rescue *PEAK1* inactivation by re-expression of WT (or mutant) PEAK1. This adds confidence that our system represents a valid model of PEAK1 KO. We hope the reviewer will be understanding of this technical limitation encountered during the revision process.

“7. Fig s7j, why were the lungs probed ex vivo. Could in vivo signal be detected?”

The size of the tumors in these mice is much bigger than the metastasis in the lungs. When you increase the bioluminescence exposure, a signal at the lungs is detected in vivo but a much bigger signal is detected at the tumors which masks the signal detected at the lungs. Hence, lungs had to be excised and probed for ex vivo at the experiment end point.

“8. Fig s7d/e: Tumor volumes should be measured, not only IVIS signal.”

The tumor masses were measured and added, and as expected, correlate with the IVIS data (Fig S7j).

“9. The pathway model should be depicted in the main figures.”

Again, this was a limitation of the “Letter” format that was opted for the initial submission to *Nature Cell Biology*. We thank the reviewer for this comment, and we have now included the model in the main figures (Fig 10i).

REFERENCES

1. Goyette, M.A. *et al.* The Receptor Tyrosine Kinase AXL Is Required at Multiple Steps of the Metastatic Cascade during HER2-Positive Breast Cancer Progression. *Cell reports* **23**, 1476-1490 (2018).
2. Kelber, J.A. *et al.* KRas induces a Src/PEAK1/ErbB2 kinase amplification loop that drives metastatic growth and therapy resistance in pancreatic cancer. *Cancer research* **72**, 2554-2564 (2012).

Reviewers' comments:

Reviewer #1 (Remarks to the Author):

This manuscript is improved over the previous version, but a few issues remain to be resolved before the work is complete

1) The method of quantifying PLA signal seems odd (total average intensity versus number of puncta). The method of analysis does not tell you how much more PLA signal is there when an interaction is detected?

2) Figure 7 still seems a mess to me. Many of the blots are blown out/have poor exposures, and the graphs of FA lifetime have no primary data, such as the examples presented in Fig 3.

Reviewer #2 (Remarks to the Author):

The authors miss the point of the Gas6 issue. Yes, gamma-carboxylation is required for Gas6-mediated AXL activation but so is association of Gas6 with PS lipids -- indeed, the role of gamma-carboxylation is to facilitate Gas6/PS association. This has been established by multiple literature reports from a variety of laboratories. The extent to which Gas6 obtained from one source or another is gamma-carboxylated is relevant only in terms of its interaction with PS. Thus, the presence of PS lipids, such as from apoptotic cellular debris or other conditions, in the experimental assay cultures is critical in governing AXL activation, regardless of the extent of Gas6 gamma-carboxylation. Now, if one has a source of Gas6 that is more extensively gamma-carboxylated than a different source, the amount of PS lipids needed to be present in the assay environment to obtain a given level of AXL activation may be lesser, but Gas6 by itself cannot activate AXL to an appreciable degree even if extensively gamma-carboxylated.

Reviewer #3 (Remarks to the Author):

The authors have clarified my previous concerns and have addressed previous points/concerns raised by carrying out further experiments.

Reviewer #4 (Remarks to the Author):

The manuscript by Afnan Abu-Thuraia, et al investigated the signaling mechanisms mediated by AXL, one of the receptor tyrosine kinases (RTKs) that are activated during metastasis of breast cancer. Using quantitative phosphoproteomics, the authors identified PEAK1 as a critical downstream factor of the AXL activation and PEAK1 promoted the turnover of focal adhesion (FA) by recruiting the FA turnover complex to paxillin. Knockout of PEAK1 inhibited breast cancer cell metastasis, which can be rescued by wild-type PEAK1 but not mutant PEAK1 that lost interaction with the pathway.

Overall, the study reported an interesting finding on the important role of PEAK1 in mediating the activity of FA turnover upon AXL activation. However, a number of key mechanisms are not clear despite the observation of AXL-mediated phosphorylation events. In addition, the logic and study design of the overall study described in the manuscript was quite complicated and not all experiments were necessary to be included in the manuscript. Some major and minor concerns are listed below. Due to these concerns, the study is not yet ready for publication in its current form.

1. Identification of PEAK1 was achieved by overlapping the phosphoproteomics data of AXL activation with the interactome data of NEDD9. However, it is quite strange that the interaction between NEDD9 and PEAK1 could not be validated (line 265). This cast doubts on the validity of

the NEDD9 interactome analysis with Bio-ID method.

2. The mechanism of how AXL activates PEAK1 was still not clear. Since the interaction between PEAK1 and AXL was validated and then PEAK1 phosphorylation increased upon AXL activation, the authors hypothesized that PEAK1 was a direct AXL target. However, mutation of the identified tyrosine residues did not inhibit AXL-mediated PEAK1 phosphorylation. So the key question is that whether PEAK1 phosphorylation was required for its activation. It is also possible that AXL activates PEAK1 indirectly or through protein-protein interactions.

3. For validation of AXL control of PEAK1 localization at FA, the authors applied siRNA and chemical treatment to knock down or inhibit AXL activity. However, these experiments are not sufficient to demonstrate the importance of AXL kinase activity in this process. Activation of AXL and kinase dead mutant by overexpression and GAS6 treatment would be more informative to determine if PEAK1 phosphorylation by AXL plays an important role during this process.

4. The overall role of NEDD9 in PEAK1-mediated phosphorylation and activation of FA turnover is not very clear. The manuscript included Figure 4 and a large section of texts describing the validation and interaction of NEDD9 with FA. However, such information only led to the identification of PEAK1 and not quite relevant with the key findings of the current manuscript. Therefore, the inclusion of these data on NEDD9 made it difficult to follow the logic flow of the study.

5. The mechanisms on how AXL-mediated CRKII phosphorylation regulates its interaction with PXN was not clear. Since the AXL-mediated phosphorylation sites on CRKII have been identified, it would be straightforward to test if these phosphorylation events are important for CRKII interaction with PXN.

6. The importance of PEAK1-mediated phosphorylation of PXN was not clear. The authors showed that PEAK1 may be able to directly phosphorylate PXN and it was also possible that CSK through PRAG phosphorylates PXN. These possibilities were not rigorously studied and should not be included if the findings are not conclusive. It is also unclear what may be the site-specific effect of PEAK1-mediated PXN phosphorylation in FA turnover.

7. The interaction between PEAK1 and PAK1 complex was identified and the authors hypothesized that these interactions are important for FA turnover. However, other than the knockdown study, there is no further mechanistic study to prove the hypothesis.

8. The authors demonstrated the important role of PEAK1 and CRKII complex in FA turnover using 3PA PEAK1 mutant. However, the key question is whether AXL mediates the interaction between PEAK1 and CRKII.

9. Since 3PA PEAK1 mutant is the key to demonstrate the important role of activation of PEAK1 by AXL in vivo, it is important to determine if the mutation affect the interaction between PEAK1 and b-PIX/GIT1/PAK1 complex.

10. The proteomics data analysis figure (Figure 1c) is confusing. For quantitative proteomics data, please perform three biological replicates for each time point and use volcano plots to identify significantly changed hits.

Minor concern:

1. Figure 1d, how the relative abundance of phosphorylation was defined? Why was the 0 min abundance for each site not shown? The figure may be better presented as line graphs.

2. Figure 2 can be put in Supplementary material as the analysis does not contribute significantly to identification of PEAK1 and other targets.

Reviewer #5 (Remarks to the Author):

The revised manuscript by Abu-Thuraia et al., addresses both the scientific concerns and the issues raised by reviewers with the presentation of the proposal during the prior cycle of review. The manuscript addresses the potential role of an AXL/PEAK regulated signaling axis consortium in promoting cell adhesion and cell migration. The work provides a new rationale for the regulation of cell motility downstream of AXL via CrkII and NEDD9, which may be important not only in

pathologies such as cancer, but also during development. The authors have done a good job of addressing most of the prior issues. There are still a couple issues related to prior points which should be addressed:

The authors conclude that PEAK1/CrkII interaction is required for "localization" at FAs. It remains possible that PEAK1 is not responsible for localization/targeting, but acts to stabilize CrkII within the FAs. This should be addressed within the text, at a minimum.

The studies are highly dependent upon cell lines and upon Axl over-expression, and this seems partly due to a lab-constructed system. Axl is expressed in only a fraction of invasive breast cancers and it is a stretch to claim it is "highly expressed." In fact, Axl is a significant negative prognostic only in renal cancer (where PEAK1 is abundant). In other tissues, which express high levels of both PEAK1 and Axl, (such as testicles, where co-expression is the highest) Axl expression is typically lost during oncogenesis. The authors should address this dichotomy in order to provide their results in a correct context.

Minor: The presentation of the bioinformatics includes gratuitous graphics (which are generally intended for the presentation of much bigger data matrixes) (eg., Figure 2a, d). In these cases, a simple table would provide more information to the reader while occupying less space.

We would like to thank the reviewers for their expert comments on our work. You'll find below a point-by-point response.

Reviewer 1.

“This manuscript is improved over the previous version, but a few issues remain to be resolved before the work is complete

1) The method of quantifying PLA signal seems odd (total average intensity versus number of puncta). The method of analysis does not tell you how much more PLA signal is there when an interaction is detected?

2) Figure 7 still seems a mess to me. Many of the blots are blown out/have poor exposures, and the graphs of FA lifetime have no primary data, such as the examples presented in Fig 3”

We thank this reviewer for the positive comment on our revised manuscript. We now addressed the two remaining comments. First, the method of PLA quantification has been modified in all PLA panels in the paper to report the number of PLA puncta per cell. Second, following the relevant comment, we have now added primary data panels of FAs turnover (Figure 7f), like in Figure 3, to Figure 7.

Reviewer 2.

“The authors miss the point of the Gas6 issue. Yes, gamma-carboxylation is required for Gas6-mediated AXL activation but so is association of Gas6 with PS lipids -- indeed, the role of gamma-carboxylation is to facilitate Gas6/PS association. This has been established by multiple literature reports from a variety of laboratories. The extent to which Gas6 obtained from one source or another is gamma-carboxylated is relevant only in terms of its interaction with PS. Thus, the presence of PS lipids, such as from apoptotic cellular debris or other conditions, in the experimental assay cultures is critical in governing AXL activation, regardless of the extent of Gas6 gamma-carboxylation. Now, if one has a source of Gas6 that is more extensively gamma-carboxylated than a different source, the amount of PS lipids needed to be present in the assay environment to obtain a given level of AXL activation may be lesser, but Gas6 by itself cannot activate AXL to an appreciable degree even if extensively gamma-carboxylated.”

We extensively characterized the GAS6 we produce to answer the previous criticisms. We did not miss the point of the reviewer, as suggested, but instead provided compelling data that we did not perform experiments in suboptimal GAS6 concentrations for our MS studies. We still agree with this reviewer that the GAS6 produced in our system may already be coated with PS providing maximal activity as demonstrated by AKT phosphorylation (Supplemental Fig 1f). We now make an explicit statement to this effect in the manuscript (see page 6).

Reviewer 3.

“The authors have clarified my previous concerns and have addressed previous points/concerns raised by carrying out further experiments.”

We would like to thank the reviewer for positively assessing our revised manuscript.

Reviewer 4.

“The manuscript by Afnan Abu-Thuraia, et al investigated the signaling mechanisms mediated by AXL, one of the receptor tyrosine kinases (RTKs) that are activated during metastasis of breast cancer. Using quantitative phosphoproteomics, the authors identified PEAK1 as a critical downstream factor of the AXL activation and PEAK1 promoted the turnover of focal adhesion (FA) by recruiting the FA turnover complex to paxillin. Knockout of PEAK1 inhibited breast cancer cell metastasis, which can be rescued by wild-type PEAK1 but not mutant PEAK1 that lost interaction with the pathway.

Overall, the study reported an interesting finding on the important role of PEAK1 in mediating the activity of FA turnover upon AXL activation. However, a number of key mechanisms are not clear despite the observation of AXL-mediated phosphorylation events. In addition, the logic and study design of the overall study described in the manuscript was quite complicated and not all experiments were necessary to be included in the manuscript. Some major and minor concerns are listed below. Due to these concerns, the study is not yet ready for publication in its current form.”

1. Identification of PEAK1 was achieved by overlapping the phosphoproteomics data of AXL activation with the interactome data of NEDD9. However, it is quite strange that the interaction between NEDD9 and PEAK1 could not be validated (line 265). This casts doubts on the validity of the NEDD9 interactome analysis with Bio-ID method.

Not all BioID proximity interactions indicate direct protein interactions. Also, BioID has the power to capture weak and transient interactions in living cell, in complex biological compartments such as membranes and FAs, that may not be amenable to biochemical procedures such as co-IP following cell lysis. Finally, BioID can also reveal proteins that have a common membership to a larger protein complex. In that respect, we show that CRKII, a major NEDD9-binding protein, interacts with PEAK1 in this manuscript and that co-expression of CRKII along with NEDD9 and PEAK1 allows the formation of a ternary complex (Fig 6D). We are convinced of the quality of the NEDD9 BioID data since we have compared it to a large number of BioID experiments carried out in the Côté and Gingras labs.

2. The mechanism of how AXL activates PEAK1 was still not clear. Since the interaction between PEAK1 and AXL was validated and then PEAK1 phosphorylation increased upon AXL activation, the authors hypothesized that PEAK1 was a direct AXL target. However, mutation of the identified tyrosine residues did not inhibit AXL-mediated PEAK1 phosphorylation. So the key question is that whether PEAK1 phosphorylation was required for its activation. It is also possible that AXL activates PEAK1 indirectly or through protein-protein interactions.

The starting point for these studies is that we identified 5 pTyr sites on PEAK1 in our unbiased phosphoscreens of GAS6 treated cells. These assays suggest that AXL itself, or one or more of its downstream tyrosine kinase targets, phosphorylate PEAK1. As stated by this reviewer, we provide evidence that AXL itself can phosphorylate PEAK1. We initially focused on Y531 since bioinformatics analyses by PhosphoNET predicted it as the top AXL candidate site. However, we found that a PEAK1 Y531F mutant was still able to be phosphorylated by AXL, which did not really surprise us given the MS results. The next step will be to carry out a full structure/function study of the 5 pTyr sites that may require the generation of a large panel of single mutants or compound mutations (double, triple, quadruple or quintuple) to identify the relevant sites. Finally, since PEAK1 is a pseudokinase, it is presently unknown what “...whether

PEAK1 phosphorylation was required for its activation” and “...it is also possible that AXL activates PEAK1 indirectly” would mean. However, the general point of the reviewer is well taken and we have now added some information in the discussion on possible pTyr-independent or protein-protein interactions-dependent mechanisms.

3. For validation of AXL control of PEAK1 localization at FA, the authors applied siRNA and chemical treatment to knock down or inhibit AXL activity. However, these experiments are not sufficient to demonstrate the importance of AXL kinase activity in this process. Activation of AXL and kinase dead mutant by overexpression and GAS6 treatment would be more informative to determine if PEAK1 phosphorylation by AXL plays an important role during this process.

We appreciate this comment. We have performed a PLA experiment to address PEAK1 localization at PXN-FAs in MDA-MB-231. This experiment was carried under conditions of GAS6 treatment, R428-mediated inhibition of AXL, and combined GAS6 and R428 treatments. Collectively, these assays further confirm our conclusion that an increase in AXL catalytic activity leads to an increase in PEAK1 localization at PXN-FAs, as shown in Figure 5d, e.

4. The overall role of NEDD9 in PEAK1-mediated phosphorylation and activation of FA turnover is not very clear. The manuscript included Figure 4 and a large section of texts describing the validation and interaction of NEDD9 with FA. However, such information only led to the identification of PEAK1 and not quite relevant with the key findings of the current manuscript. Therefore, the inclusion of these data on NEDD9 made it difficult to follow the logic flow of the study.

We thank this reviewer for this comment. While we appreciate the comment, we strongly believe that it was important to characterize NEDD9, since we identified it as a major substrate of AXL. It is indeed this data that guided us to conduct BioID on NEDD9 and to ultimately converge on PEAK1 as a protein appearing in all pMS and BioID screens. Nevertheless, we made some text changes to shorten this section and re-focus it to transition better to PEAK1.

5. The mechanisms on how AXL-mediated CRKII phosphorylation regulates its interaction with PXN was not clear. Since the AXL-mediated phosphorylation sites on CRKII have been identified, it would be straightforward to test if these phosphorylation events are important for CRKII interaction with PXN.

We are pleased that this reviewer explored our MS data. Indeed, GAS6 stimulation led to the phosphorylation on CRKII (Y221, Y251) and CRKL (Y207, Y251). As this reviewer can imagine, we could not follow-up experimentally on all sites identified in the screen. We have now removed the statement that CRKII phosphorylation contributes to binding to PXN. These are certainly interesting future lines of investigations that we hope our large datasets will stimulate such studies (our lab and others).

6. The importance of PEAK1-mediated phosphorylation of PXN was not clear. The authors showed that PEAK1 may be able to directly phosphorylate PXN and it was also possible that CSK through PRAG phosphorylates PXN. These possibilities were not rigorously studied and should not be included if the findings are not conclusive. It is also unclear what may be the site-specific effect of PEAK1-mediated PXN phosphorylation in FA turnover.

We do not suggest that PEAK1 directly phosphorylates PXN. PEAK1 is a *bona fide* pseudokinase. However, we have compelling evidence that depletion of PEAK1 abrogates PXN

phosphorylation. We also show that PEAK1 co-precipitates with CSK and that co-expression of PEAK1/CSK enhances PXN phosphorylation. The reviewer is correct that the exact mechanism is not completely understood from our work. In response to this insightful comment, we have now moved the PRAG1 arguments from the Results to the Discussion section.

7. The interaction between PEAK1 and PAK1 complex was identified and the authors hypothesized that these interactions are important for FA turnover. However, other than the knockdown study, there is no further mechanistic study to prove the hypothesis.

This comment was addressed together with the comments raised in the related point #9 below.

8. The authors demonstrated the important role of PEAK1 and CRKII complex in FA turnover using 3PA PEAK1 mutant. However, the key question is whether AXL mediates the interaction between PEAK1 and CRKII.

We appreciate this comment. We have addressed this question by co-immunoprecipitation of CRKII with PEAK1 in conditions where wildtype AXL was expressed and in other conditions where kinase dead AXL was expressed and found that CRKII/PEAK1 interactions was only mediated upon wildtype AXL expression, suggesting that AXL kinase activity mediates PEAK1/CRKII interaction, shown in Figure 6d.

9. Since 3PA PEAK1 mutant is the key to demonstrate the important role of activation of PEAK1 by AXL in vivo, it is important to determine if the mutation affect the interaction between PEAK1 and b-PIX/GIT1/PAK1 complex.

This is a valid point to address and we thank the reviewer for bringing this up to our attention. We now show by co-immunoprecipitations, the levels of GIT1/ β PIX-binding to PEAK1 is decreased upon expression of PEAK1 3PA mutant as shown in Figure S7l, m. This might explain why FA size and number were increased upon expression of PEAK1 3PA mutant in PEAK1 KO cells, as shown in Figure 10f-h.

10. The proteomics data analysis figure (Figure 1c) is confusing. For quantitative proteomics data, please perform three biological replicates for each time point and use volcano plots to identify significantly changed hits.

We appreciate this comment. We have performed our SILAC experiments in duplicates (n=2 for pS/T enrichment and n=2 for pY enrichment for each timepoint). Hence, we cannot perform any t-tests and generate p-values from those duplicates. That's why we opted to present the localization probabilities instead, which is calculated by MaxQuant using the average of the phosphorylation fold change of the two duplicates (pS/T and pY).

In addition, the use of SILAC approach has been published previously in respected journals as one biological replicate or in duplicates since it represents an accurate quantitation of protein expression and site-specific phosphorylation. Here are some examples of published papers with one or two replicates performed for SILAC: (Galan JA et al. *PNAS*, 2014), (Gonneaud A et al. *Sci Rep*, 2016) and (Zhao F et al. *Proteomics*, 2016).

Minor concern:

1. Figure 1d, how the relative abundance of phosphorylation was defined? Why was the 0 min abundance for each site not shown? The figure may be better presented as line graphs.

Relative abundance was defined based on the ProHits-viz tool (<https://prohits-viz.lunenfeld.ca>) generated by co-author Anne-Claude Gingras' team to generate dot plots which can depict the relative differences between values picked up for a certain site at different conditions. For each site, the values used to generate the relative abundance was the average SILAC ratios H/L for the two replicates performed.

2. Figure 2 can be put in Supplementary material as the analysis does not contribute significantly to identification of PEAK1 and other targets.

One goal of our work is to reveal the first phospho-MS analyses of AXL signalling. While we do not follow up on all biological processes, we believe Figure 2 provides a quantitative overview of the pathways and biological processes regulated by AXL signalling.

Reviewer 5.

“The revised manuscript by Abu-Thuraia et al., addresses both the scientific concerns and the issues raised by reviewers with the presentation of the proposal during the prior cycle of review. The manuscript addresses the potential role of an AXL/PEAK regulated signaling axis consortium in promoting cell adhesion and cell migration. The work provides a new rationale for the regulation of cell motility downstream of AXL via CrkII and NEDD9, which may be important not only in pathologies such as cancer, but also during development. The authors have done a good job of addressing most of the prior issues. There are still a couple issues related to prior points which should be addressed:

The authors conclude that PEAK1/CrkII interaction is required for “localization” at FAs. It remains possible that PEAK1 is not responsible for localization/targeting, but acts to stabilize CrkII within the FAs. This should be addressed within the text, at a minimum.

We thank this reviewer for a critical assessment of our work and of previous reviews. We agree with this reviewer and have included a sentence suggesting a possible role for PEAK1 in stabilizing CRK into FAs.

The studies are highly dependent upon cell lines and upon Axl over-expression, and this seems partly due to a lab-constructed system. Axl is expressed in only a fraction of invasive breast cancers and it is a stretch to claim it is “highly expressed.” In fact, Axl is a significant negative prognostic only in renal cancer (where PEAK1 is abundant). In other tissues, which express high levels of both PEAK1 and Axl, (such as testicles, where co-expression is the highest) Axl expression is typically lost during oncogenesis. The authors should address this dichotomy in order to provide their results in a correct context.”

We believe that this reviewer misinterpreted the expression approach of AXL. All experiments were conducted on endogenous AXL in MDA-MB-231 and HS578T cells, which are well-established to robustly express AXL at the protein level. In this study, we downregulated AXL expression by siRNA or inhibited its kinase activity with the pharmacological inhibitor R428.

But no functional experiment relied on overexpression. We did conduct overexpression experiments in HEK293 cells with AXL to assess protein complexes formation by co-IP.

Our laboratory also showed that AXL is expressed in a fraction of patient samples across all breast cancer subtype (Goyette MA et al. *Cell Reports*, 2018). While AXL was believed not to be expressed in HER2+ cancers, we found a correlation of AXL expression with the HER2+ cancers that also display EMT gene signatures and that this association negatively correlates with survival. We also note in our introduction that AXL is not express in all breast cancers. We now include an additional comment in the discussion. We feel that broadly discussing other cancer types is beyond the scope of this manuscript.

Minor: The presentation of the bioinformatics includes gratuitous graphics (which are generally intended for the presentation of much bigger data matrixes) (eg., Figure 2a, d). In these cases, a simple table would provide more information to the reader while occupying less space.

We respectfully disagree with this comment. We provide all the raw data from our phospho-MS analyses in the supplemental table. We believe that our representation provides a useful breakdown of the data into biological processes that would not be achievable by a table.

REVIEWERS' COMMENTS:

Reviewer #1 (Remarks to the Author):

The authors have satisfactorily addressed all of the points necessary for publication.

Reviewer #2 (Remarks to the Author):

The authors state that now on page 6 that there is minimal phosphatidylserine due to apoptotic cells in their assay. Nonetheless, it remains imperative that their Gas6-His be complexed with phosphatidylserine in order to activate AXL. Thus, the authors need to explain where the phosphatidylserine is coming from in order to account for their Gas6-His mediated AXL phosphorylation.

Reviewer #4 (Remarks to the Author):

The reviewer appreciates the efforts of the authors. The revised manuscripts with new experiments have addressed most of the reviewer's concerns. Although some caveats remain, including the functional relevance of PEAK1 phosphorylation and duplicate analysis of SILAC quantification, this reviewer agrees that they are not essential to this study and the major conclusions are supported by a large body of evidence. Overall, the work identified an important signaling pathway with functional significance in cancer metastasis and it is recommended for publication.

Reviewer #1

The authors have satisfactorily addressed all of the points necessary for publication.

Thank you for your generous time to review our manuscript again.

Reviewer #2

The authors state that now on page 6 that there is minimal phosphatidylserine due to apoptotic cells in their assay. Nonetheless, it remains imperative that their Gas6-His be complexed with phosphatidylserine in order to activate AXL. Thus, the authors need to explain where the phosphatidylserine is coming from in order to account for their Gas6-His mediated AXL phosphorylation.

We have added text (underlined) as requested by the reviewer: “This suggests that GAS6-His likely coated with phosphatidylserine vesicles in the conditioned media or that the Hs578T cells exposed some phosphatidylserine at their surface in a physiological manner.”

Reviewer #4

The reviewer appreciates the efforts of the authors. The revised manuscripts with new experiments have addressed most of the reviewer’s concerns. Although some caveats remain, including the functional relevance of PEAK1 phosphorylation and duplicate analysis of SILAC quantification, this reviewer agrees that they are not essential to this study and the major conclusions are supported by a large body of evidence. Overall, the work identified an important signaling pathway with functional significance in cancer metastasis and it is recommended for publication.

We appreciate that the reviewer appreciated the overall significance of our work.